# Probing spectral features of quantum many-body systems with quantum simulators

Jinzhao Sun ®[1,2]✉, Lucia Vilchez-Estevez ®[1]✉, Vlatko Vedral[1], Andrew T. Boothroyd[1] & M. S. Kim ®[2]

The efficient probing of spectral features is important for characterising and understanding the structure and dynamics of quantum materials. In this work, we establish a framework for probing the excitation spectrum of quantum many-body systems with quantum simulators. Our approach effectively realises a spectral detector by processing the dynamics of observables with time intervals drawn from a defined probability distribution, which only requires native time evolution governed by the Hamiltonian without ancilla. The critical element of our method is the engineered emergence of frequency resonance such that the excitation spectrum can be probed. We show that the time complexity for transition energy estimation has a logarithmic dependence on simulation accuracy and how such observation can be guaranteed in certain many-body systems. We discuss the noise robustness of our spectroscopic method and show that the total running time maintains polynomial dependence on accuracy in the presence of device noise. We further numerically test the error dependence and the scalability of our method for lattice models. We present simulation results for the spectral features of typical quantum systems, either gapped or gapless, including quantum spins, fermions and bosons. We demonstrate how excitation spectra of spin-lattice models can be probed experimentally with IBM quantum devices.

Estimating spectral features of quantum many-body systems has attracted great attention in condensed matter physics and quantum chemistry. To achieve this task, various experimental and theoretical techniques have been developed, such as spectroscopy techniques[1-7] and quantum simulation either by engineering controlled quantum devices[8-16] or executing quantum algorithms[17-20] such as quantum phase estimation and variational algorithms. However, probing the behaviour of complex quantum many-body systems remains a challenge, which demands substantial resources for both approaches. For instance, a real probe by neutron spectroscopy requires access to large-scale facilities with high-intensity neutron beams, while quantum computation of eigenenergies typically requires controlled operations with a long coherence time[17,18]. Efficient estimation of spectral

properties has become a topic of increasing interest in this noisy intermediate-scale quantum era[21].

A potential solution to efficient spectral property estimation is to extract the spectral information from the dynamics of observables, rather than relying on real probes such as scattering spectroscopy, or direct computation of eigenenergies. This approach capitalises on the basics in quantum mechanics that spectral information is naturally carried by the observable's dynamics[10,20,22-26]. In a solid system with translation invariance, for instance, the dynamic structure factor, which can be probed in spectroscopy experiments[7,26], reaches its local maximum when both the energy and momentum selection rules are satisfied. Therefore, the energy dispersion can be inferred by tracking the peak of intensities in the energy excitation spectrum. Inspired by

[1]Clarendon Laboratory, University of Oxford, Parks Road, Oxford, United Kingdom. [2]Blackett Laboratory, Imperial College London, London, United Kingdom. ✉e-mail: jinzhao.sun.phys@gmail.com; lucia.vilchezestevez@physics.ox.ac.uk

spectroscopy, a straightforward way to detect spectral features of model systems is by directly simulating spectroscopy[16,20,26–28] using quantum computers, which often requires the measurement of an unequal-time correlator on a thermal state with long-time evolution. Another similar idea is to extract spectral information by post-processing the time-dependent signals[10,22–25,29–32] which is usually ancilla-free. For example, Zintchenko and Wiebe[30] proposed to estimate the spectral gap by using Bayesian inference of the measurement outcomes generated by applying random unitaries (see other similar works[31,32]). Wang et al.[33,34] and Stenger et al.[28] proposed detecting energy differences by measuring the dynamical response of a quantum system when coupled to a probe qubit. Recently, Chan et al. proposed to perform the Fourier transform of observable's dynamics and innovatively use shadows[35] to estimate many observables simultaneously[25] (see other relevant works[10,22–24,36]). Nevertheless, methods that extract spectral information from dynamics face challenges in achieving high-precision results with short-time dynamics. In addition, it is generally hard to design appropriate spectroscopic protocols for many-body systems. A pressing question is whether we can probe the spectral features and obtain high-precision results with fewer quantum resources.

In this work, we introduce a spectroscopic method that links spectroscopy techniques and quantum simulation while addressing the above challenges in probing transition energies and excitation spectra. We show how a spectral detector can be effectively realised by processing the dynamics of observables with time intervals drawn from a defined probability distribution. The maximum time complexity is found to be logarithmic in precision under assumptions similar to those used in eigenenergy estimation[37–44], which enables high-precision simulation with short-time dynamics. The essential requirement of the spectroscopic method is a nonvanishing observation of the target transitions between eigenstates, which depends on the initial state and the observable. We illustrate how this nonvanishing observation can be guaranteed in some representative many-body systems with quasi-particle number conservation. In terms of practical implementation, our method only requires the realisation of time evolution $e^{-iHt}$, in contrast to existing algorithms for eigenenergy estimation or simulated spectroscopy, which rely on controlled unitaries as a subroutine, either using Hamiltonian simulation[27,37–40] or variational algorithms[16,20]. We further analyse the noise robustness on spectral property estimation, including both coherent and incoherent noise. In the presence of device noise, the total running time maintains polynomial in inverse precision, showing advantages over existing approaches such as the standard quantum phase estimation and Fourier transform of the time signals[22,24,25,36]. We test the performance of the error mitigation strategy and the spectroscopic protocol by including the device and statistical noise in the numerical simulation. Finally, we investigate our method in cases of quantum spins, bosons, and fermions by numerics and simulation on IBM quantum devices. We show how transition energies and excitation spectra of spin-lattice models can be probed with IBM quantum devices with 13 qubits and over 350 two-qubit gates.

## Results

### Motivation

The critical element in spectroscopy is the emergence of frequency excitations (and momentum excitations in translationally invariant systems) corresponding to the elementary excitations between eigenstates, which enables us to probe the transition energies as well as excitation spectra. Nevertheless, there are several constraints to conventional spectroscopy approaches, including (1) the perturbed system $\rho$ is in an equilibrium state $[\rho, H] = 0$, (2) the perturbation is weak and the linear response theory holds. As there is no coherence of the initial state, ($\rho$ is diagonal in the eigenbases $|n\rangle$ of the Hamiltonian, $\langle n|\rho|n'\rangle = \delta_{nn'} e^{-\beta E_n}$), we can only probe properties in the equilibrium phase.

The dynamical structure factor, which can be obtained from spectroscopy experiments, reflects the energy resonance between eigenstates $|n'\rangle$ and $|n\rangle$ and can be expressed as $S(\omega) = \sum_{n,n'} \rho^{nn} A_{n',n} \delta(E_{n'} - E_n - \omega)$ with $A_{n',n} = \langle n|\hat{O}_1^\dagger|n'\rangle\langle n'|\hat{O}_2|n\rangle$ and observables $\hat{O}_1$ and $\hat{O}_2$ (e.g. spin operators). The dynamical structure factor $S(\omega)$ is a Fourier transform of a two-point unequal-time correlation function $C(t) = \text{Tr}(\rho \hat{O}_1^\dagger(t)\hat{O}_2)$ in the Heisenberg picture on the equilibrium state $\rho$, $S(\omega) = \int_{-\infty}^{+\infty} C(t)e^{i\omega t}dt$. A straightforward way to detect the transition energy is by directly simulating the spectroscopy process, as studied in refs. [16,26–28]. However, it is less efficient since the time complexity for realising the spectral function $S(\omega)$ is large and an ancillary qubit is required for measuring $C(t)$ when using the Hadamard-test circuit. In addition, we need to prepare a thermal state $\rho$ and can detect the equilibrium properties only. This raises the question of whether we can reduce the simulation time while maintaining simulation accuracy.

### Framework

Here we develop a framework for estimating transition energies between the eigenstates of a quantum many-body system. Let us consider a quantum operation $\mathcal{G}(\rho, \omega) = \sum_{n,n' \geq 0} \rho^{n'n}|n'\rangle\langle n|p_\tau(E_{n'} - E_n - \omega)$, where $\rho^{n'n} := \langle n'|\rho|n\rangle$ and a function $p_\tau(\cdot)$ that selects the energy difference between eigenstates $|n'\rangle$ and $|n\rangle$ is introduced, for instance, the Gaussian function $p_\tau(\omega) = \exp(-\tau^2\omega^2)$. With a properly selected observable $\hat{O}$, we can obtain the measurement outcome $G(\omega) = \text{Tr}[\mathcal{G}(\rho, \omega)\hat{O}]$ which can be expressed by

$$G(\omega) := \sum_{n, n' \geq 0} \Gamma_{n',n} p(\tau(E_{n'} - E_n - \omega)), \tag{1}$$

where $\Gamma_{n',n} := \rho^{n'n}\langle n|\hat{O}|n'\rangle$ represents the state-and-observable dependent coherence, yet is time-independent; $\Gamma_{n',n}$ can also be regarded a spectral weight associated with the initial state and observable. Here, we first arrange that $p_\tau(\omega) = p(\tau\omega)$, such that $\tau$ is coupled with $\omega$. The quantity $G(\omega)$ contains the information on transition energies $\Delta_{n',n} := E_{n'} - E_n$. Specifically, given a proper $p(\cdot)$ and a large coherence $\Gamma_{n',n}$ between $|n'\rangle$ and $|n\rangle$, $G(\omega)$ takes its local maximum when $\omega$ approaches $\Delta_{n',n}$, and thus serves as a spectral detector. $G(\omega)$ characterises similar features as that of the dynamic structure factor, but its realisation will require fewer quantum resources than the dynamic structure factor.

A question is how to effectively implement the quantum operation and estimate $G(\omega)$ in Eq. (1). A natural idea is to effectively realise $G(\omega)$ by real-time dynamics, which is usually easy to implement on quantum simulators. To do so, we consider a Fourier transform of $G(\omega)$. Specifically, the dual form of $p$ via its Fourier transform is given by $\tilde{g}(t) := \int_{-\infty}^{+\infty} p(\tau\omega)e^{i\tau\omega t}d(\tau\omega)$, and its inverse form $p(\tau\omega) = \frac{1}{2\pi}\int_{-\infty}^{+\infty} \tilde{g}(t)e^{-i\tau\omega t}dt$. Consider the normalised function $g(t) = |\tilde{g}(t)|/c$ with the normalisation factor $c := \int_{\infty}^{\infty} |\tilde{g}(t)|dt$ and phase $e^{i\theta_t} := \tilde{g}(t)/|\tilde{g}(t)|$, and we have $g(t) = \frac{1}{c}\int_{-\infty}^{+\infty} p(\tau\omega)e^{-i\theta_t}e^{i\tau\omega t}d(\tau\omega)$ and its dual form $p(\tau\omega) = \frac{c}{2\pi}\int_{-\infty}^{+\infty} g(t)e^{i\theta_t}e^{-i\tau\omega t}dt$. Plugging the Fourier transform of $p$ into Eq. (1), the spectral detector takes the form of

$$G(\omega) = \frac{c}{2\pi}\int_{\infty}^{\infty} G(\tau t)g(t)e^{i\theta_t}e^{i\tau\omega t}dt \tag{2}$$

with $G(t) := \text{Tr}[\hat{O}\rho(t)]$ in the Schrödinger picture. Eq. (2) indicates that we can first obtain $\text{Tr}[\hat{O}\rho(\tau t)]$ by measuring $\hat{O}$ on the time-evolved state at time $\tau t$, and then use Eq. (2) to obtain $G(\omega)$. Since $g(t)$ is normalised and hence can be regarded as a probability distribution, a single-shot estimator of $\hat{G}(\omega)$ takes the form of

$$\hat{G}(\omega) = \frac{c}{2\pi}\hat{o}(\tau t_i)e^{i\theta_{t_i}}e^{i\tau\omega t_i}, \tag{3}$$

where $t_i$ is sampled from the distribution $g(t)$ which is $\tau$-independent, and $\hat{o}(\tau t_i)$ is an unbiased estimate of $\mathrm{Tr}\,[\hat{O}(\tau t)\rho]$. One can verify that $\hat{G}(\omega)$ is an unbiased estimator of $G(\omega)$, $G(\omega) = \mathbb{E}\hat{G}(\omega)$, where the average is taken over the probability distribution $g(t)$. Here, we choose to treat $\tau\omega$ as a whole in Eq. (2) when performing the Fourier transform. As such, $g(\cdot)$ is a $\tau$-independent probability distribution, and the total time length for evaluating $G(\omega)$ is $\tau t$, which is extended by a factor $\tau$ compared to original Fourier transform. An advantage of this treatment is that it enables a simple evaluation of the resource requirements using different functions $p$ within a unified framework, rather than a case-by-case analysis. Alternatively, we can treat $\tau$ as a variable (in $p_\tau(\cdot)$) that is independent of $\omega$, and hence will not be Fourier-transformed. These two ways are proven to be equivalent in Methods.

There are two necessary requirements for inferring the transition energy $\Delta_{n',n}$ using the spectral detector in Eq. (1): (1) a sufficiently large state-and-observable coherence $\Gamma_{n',n}$, and (2) a proper function $p(\omega)$ (or equivalently its dual form $g(t)$) that ensures that $\Delta_{n',n}$ can be distinguished from other transition energies. The coherence $\Gamma_{n',n}$ is time-independent yet dependent on the state and the chosen observable. In the following sections, we first discuss the selection of the initial state and observables in order to satisfy the first condition. We then show that by choosing the Gaussian function $p(\omega)$, our method only requires short-time dynamics to achieve high-precision energy estimation.

In a concurrent work[43], Yang et al. developed a similar method for evaluating the energy gap by introducing a so-called tuning parameter to increase the convergence rate, although the choice of the tuning parameter is not discussed and is fixed in numerics. Within our framework, the tuning parameter can be regarded as a separate parameter that is irrelevant to the Fourier transform.

## Spectroscopic protocol

Now, we discuss the selection of the initial state and observables in several representative quantum systems such that $\Gamma_{n',n}$ is nonvanishing. Several works have discussed how to probe the excitation spectra by engineering the controllable quantum system[8,23,24,44–48]. The basis of the spectroscopic protocols is that the initial state is populated by a superposition of low-lying excited states, which could be expressed as $|\psi_0\rangle = \sum_j a_j |j\rangle$ where $|j\rangle$ is an eigenstate of $H$. The initial state, generated by a global or local operation $\hat{B}$, could be formally expressed as $|\psi_0\rangle = b^{-\frac{1}{2}}\hat{B}|0\rangle$. Here $b := \langle\psi_0|\hat{B}^\dagger\hat{B}|\psi_0\rangle$ is the normalisation factor, and $|0\rangle$ denotes the ground state of $H$. The observation in relation to the transition between the excited state and the ground state is $\langle n|\hat{O}|0\rangle = \sum_j a_j b^{\frac{1}{2}}\langle n|\hat{O}\hat{B}^{-1}|j\rangle$, and the state-and-observable coherence can be nonzero by choosing an appropriate $\hat{B}$.

In solid systems, translation invariance is usually conserved and the Hamiltonian satisfies $[H, \hat{\mathbf{P}}] = 0$, where $\hat{\mathbf{P}}$ is the total momentum operator. Each eigenstate $|n\rangle$ has a well-defined momentum of $\mathbf{p}_n$, $\hat{\mathbf{P}}|n\rangle = \mathbf{p}_n|n\rangle$. Suppose we choose the observable at position $\mathbf{x}$, $\hat{O}(\mathbf{x}) = e^{-i\hat{\mathbf{P}}\cdot\mathbf{x}}\hat{O}e^{i\hat{\mathbf{P}}\cdot\mathbf{x}}$ with $\hat{O} := \hat{O}(\mathbf{0})$. Taking a space Fourier transform of $G_x(\omega)$ in Eq. (1) with the observable $\hat{O}(\mathbf{x})$, $G_k(\omega) = \int d\mathbf{x}\, e^{-i\mathbf{k}\mathbf{x}}G_x(\omega)$, we have

$$G_k(\omega) = 2\pi \sum_{n,n'=0} \Gamma_{n',n}\, p(\tau(E_{n'} - E_n - \omega))\delta(\mathbf{p}_{n'} - \mathbf{p}_n - \mathbf{k}), \quad (4)$$

when the system size reaches infinity and the translation invariance of the initial state is broken; for instance, the translation invariance of the state after applying a local operation to a single site is broken. The above equation indicates that translation invariance imposes selection rules of both energy and momentum for transition between eigenstates. This is the key element in spectroscopy detection, where elementary excitations between eigenstates emerge when the selection rules of energy and momentum are both satisfied.

Although in general a large coherence cannot be guaranteed (as this problem is quantumly hard; see Discussion), there are certain cases where we can manipulate the system to meet this requirement and allow for probing specific types of excitations. In a weakly coupled system, for instance, the particle excitations induced by perturbations are restricted to a manifold of single-particle excitations, as discussed in refs. 24,44. In this limit, an excited state could be understood as a single particle (or quasiparticle) excitation above the ground state $|0\rangle$, $|n\rangle = \hat{\gamma}_\mathbf{q}^\dagger|0\rangle$, carrying momentum $\mathbf{q}$, where $\hat{\gamma}_\mathbf{q}^\dagger$ is the creation operator of a particle with momentum $\mathbf{q}$. Note that $\hat{\gamma}^\dagger$ does not have to be the same creation operator in the Hamiltonian and could be either the creation operator of a particle or quasiparticle. The excitation generated from the ground state can be observed by choosing $\hat{O} = \sum_\mathbf{p} A_\mathbf{p}\hat{\gamma}_\mathbf{p} + A_\mathbf{p}^*\hat{\gamma}_\mathbf{p}^\dagger$, and we have $\langle 0|\hat{O}|n\rangle = A_\mathbf{q}\delta_{\mathbf{qp}}$. To probe the single particle excitation above the ground state with energy $E_n - E_0$, we may prepare the state containing a small perturbation with momentum $\mathbf{q}$ as $|\psi_0\rangle \approx |0\rangle + \beta|n\rangle$ where $\beta$ is a small constant (see ref. 49 and Supplementary Section III). This choice of state and observable enables a nonzero observation as $\Gamma_{n,0} = \beta A_\mathbf{q}$, and the excitation spectrum can thus be observed.

We give some comments on more general cases. Let us consider the Hamiltonian which conserves either the particle or quasiparticle number. We denote the eigenstates of $H$ as $|n\rangle$ and the vacuum state $|0\rangle$, and suppose the system has $L$ modes (either in real space or momentum space). Any single-particle state $\hat{\gamma}_\mathbf{p}^\dagger|0\rangle$, which is generated by creating a particle at the $\mathbf{p}$th mode, could be decomposed into the basis of $|n\rangle$, and the decomposition coefficient is denoted as $\langle n|\hat{\gamma}_\mathbf{p}^\dagger|0\rangle := c_{n,\mathbf{p}}$. Given that the quasiparticle picture holds, we can prepare the initial state $|\psi_0\rangle = \frac{1}{\sqrt{1+\beta^2}}(1 + \beta\hat{\gamma}_\mathbf{p}^\dagger)|0\rangle$. Then the initial state coherence is $\rho^{n0} = \beta\langle n|\hat{\gamma}_\mathbf{p}^\dagger|0\rangle/(1+\beta^2) = \beta c_{n,\mathbf{p}}/(1+\beta^2)$. The transition amplitude observed by the annihilation operator on the $\mathbf{p}'$ th mode is $\langle 0|\hat{\gamma}_{\mathbf{p}'}|n\rangle = c_{\mathbf{p}',n}^*$. Therefore the coherence observed by $\hat{O}$ is given by

$$\Gamma_{n,0} = \sum_{\mathbf{p}'} \frac{\beta A_\mathbf{p}'}{1+\beta^2} c_{\mathbf{p}',n}^* c_{n,\mathbf{p}}. \quad (5)$$

A similar fashion can be used to probe the transition energy between the excited states $|n\rangle$ and $|n'\rangle$ with the same particle number. We can prepare the initial state by creating two particles as $|\psi_0\rangle = \frac{1}{1+\beta^2}(1 + \beta\hat{\gamma}_\mathbf{p}^\dagger)(1 + \beta\hat{\gamma}_{\mathbf{p}'}^\dagger)|0\rangle$, and choose the observable $\hat{O} = \sum_{\mathbf{p},\mathbf{p}'} A_{\mathbf{p},\mathbf{p}'}\hat{\gamma}_\mathbf{p}^\dagger\hat{\gamma}_{\mathbf{p}'}$ which conserves the particle number. More detailed derivations of the coherence $\Gamma_{n',n}$ can be found in Methods. Since the Hamiltonian can be engineered, we can specifically engineer quantum devices to detect the energy excitation spectra of various quantum systems. Typical transitions could be observed by preparing initial states consisting of the desired superposition (see Methods and Supplementary Information for discussions).

When the target quantum system's eigenstates can be labelled, we could track this label and detect the energy difference between these two states. This condition is satisfied when the quasiparticle picture holds. For example, in the case of Fermi liquids, the low-energy eigenstates are labelled by a set of quantum numbers $n_{\mathbf{p},\sigma} = 0, 1$ (i.e., the occupation numbers)[50], and we can ensure a nonzero coherence in the thermodynamic limit. This approach can thus be useful in detecting Fermi-liquid and some non-Fermi-liquid systems such as BCS types of systems. Another application of this spectroscopy approach is that we could detect if a system exhibits a non-Fermi-liquid feature and identify the transitions from a Fermi-liquid to a non-Fermi-liquid phase.

In addition to the above example, spectroscopy protocols have demonstrated that excitations can be effectively created in many quantum systems through numerical simulations[23,24,51–53] or analogue quantum simulations[8,22,44] in cases of the Bose-Hubbard model[22,23], spin

chains[8,13,44], and disordered systems[47]. It is worth mentioning how our method differs from these spectroscopy protocols. In the first place, the aim here is energy extraction instead of the experimental observables in spectroscopy experiments. In neutron spectroscopy experiments, for example, the externally injected neutrons act as a weak perturbation, which can probe the intrinsic properties of materials in the equilibrium state. In global or local quench spectroscopy, the eigenstates are assumed to be nearly unchanged after the quench, which is similar to that in spectroscopy experiments, although the state will be driven out of equilibrium. Our method does not put such constraints on the eigenstates and is applicable for probing more general quantum many-body systems given that the coherence associated with the excitation is nonvanishing. Moreover, quench spectroscopy is essentially limited to analogue simulations, and thus cannot be used to probe the unequal-time correlation function or systems without particle conservation. Our framework enables a direct extension to the probing of higher-order time correlation functions. For example, we can probe the nonlinear spectroscopic features[6,54] of the target system by applying perturbations multiple times (at time $t_i$ for different $i$) and obtaining the corresponding higher-order time correlation functions. The advantages over analogue quantum simulations are discussed in Supplementary Section III. Recently, ref. [55] presents a way of simulating the correlation functions in a linear response framework. Regarding the simulation of the spectroscopic features, this could be regarded as a special case of our method when taking the filter operator to be an identity and the comparisons can be found in Supplementary Information.

### Error analysis and resource requirement

We then discuss the computational complexity in transition energy estimation. In order to observe $\Delta_{n',n}$, the coherence $\Gamma_{n',n}$ associated with the transition is assumed to be nonvanishing, i.e. $\Gamma_{n',n}$ can be lower-bounded by a polynomial of the inverse of system size $N$ as $\Gamma_{n'n} \geq \Omega(\text{Poly}(\frac{1}{N}))$. This condition can be satisfied in certain cases as discussed above. For instance, in the above particle-number conserved case, the initial state is populated by a collection of low-lying excited states, in which $\Gamma_{n,0}$ is nonvanishing while $\Gamma_{n,n' \geq 1}$ is of higher order. To simplify the discussion, we will sort $\Delta_{n',n}$ in ascending order hereafter, and denote the ordered transition energies as $\Delta_j$ and the spectral gap difference as $\gamma_j := \min(\Delta_{j+1} - \Delta_j, \Delta_j - \Delta_{j-1})$. It is worth noting that in the excitation spectrum analysis, the index $j$ of the gap difference runs over the possible allowed excitations which should satisfy the momentum and coherence selection rules.

The objective is to estimate the transition energy $\Delta_j$ within an error $\varepsilon$, i.e., $|\hat{\Delta}_j - \Delta_j| \leq \varepsilon$. From now on, this is converted into an estimation problem. Intuitively, $\Delta_j$ can be determined by searching peaks of the absolute value of $G(\omega)$ in the frequency domain $\omega$. To see this point, let us rewrite $G(\omega)$ as

$$G(\omega) = \Gamma_j p(\tau(\Delta_j - \omega)) + \sum_{i \neq j} \Gamma_i p(\tau(\Delta_i - \omega)). \quad (6)$$

Given a large $\tau$, the first term will be dominant in $G(\omega)$, and thus $G(\omega)$ is close to the peak value $\Gamma_j$ only if $\omega$ is close to $\Delta_j$. In addition, in the vicinity of $\Delta_j$, i.e. $\omega \in [a_L, a_R]$, we have $\partial^2 G(\omega)/\partial^2 \omega < 0$. The transition energy can thus be estimated by finding the peak of the estimate $\hat{G}(\omega)$ concerning finite measurement, i.e., $\hat{\Delta}_j = \text{argmax}_{\omega \in [a_L, a_R]} |\hat{G}(\omega)|$. The results established in spectral filter methods can be used to analyse the simulation cost[25,30,37–39]. The total running time (maximal evolution time × sampling numbers) in ref. [37] or ref. [30] is proven to reach the Heisenberg limit for eigenenergy estimation as $\tilde{\mathcal{O}}(\varepsilon^{-1})$. Note that it is sub-optimal concerning the maximal time complexity, which is a more important metric for implementation with near-term devices due to their short coherence time. We shall see that a relatively small $\tau$ of the

order of $\log(\varepsilon^{-1})$ suffices to suppress contributions from the other transition and enables accurate estimation of $\Delta_j$, as found in the eigenenergy estimation task[39].

We illustrate the proof idea in the following and refer to Methods for details. For $\tau = \mathcal{O}(\gamma_j^{-1}\log^{\frac{1}{2}}(\varepsilon^{-1}))$, one can show that (1) $|\hat{G}(\omega) - \Gamma_j| < c_1\tau^2\varepsilon^2$, $\forall |\omega - \Delta_j| \leq 0.5\varepsilon$; (2) $|\hat{G}(\omega) - \Gamma_j| > c_2\tau^2\varepsilon^2$, $\forall |\omega - \Delta_j| \in [\varepsilon, 0.1\gamma_j]$. Here, $c_1$ and $c_2$ are some constants that are irrelevant to $\tau$ and $\varepsilon$ yet are dependent on $\Gamma_j$, and $\gamma_j$ is the gap difference (see Lemma 1 in Methods). This indicates that the distance $d = |\Gamma_j - \hat{G}(\omega)|$ is modulated by the estimation error $|\omega - \Delta_j|$, and consequently, $\Delta_j$ can be distinguished from the other transitions when $\omega$ approaches $\Delta_j$. It is assumed that $\Gamma_j$ is nonvanishing, otherwise the peak will not appear. Given the theoretical guarantees, we first get an estimate of $G(\omega)$ by Eq. (3), and $\Delta_j$ is then determined by finding the peak of $\hat{G}(\omega)$ over frequency $\omega$. Note that calculating $\hat{G}(\omega)$ as a function of $\omega$ is a task involving purely classical computing, and does not cost any quantum resources.

A remaining issue is the error from the finite cutoff when evaluating the integral in Eq. (2) with the integral range from $(-\infty, +\infty)$ to $[-T, +T]$. We highlight that our framework allows for a straightforward evaluation of the truncation error and the requirement for $T$. One can easily find that $T = \mathcal{O}(\log^{\frac{1}{2}}(\varepsilon^{-1}))$ suffices to guarantee the cutoff error below $\varepsilon$. Therefore, a finite cutoff only contributes to a logarithmic factor to the circuit complexity. The algorithmic complexity concerning a finite $\tau$, a finite cutoff $T$ for the integral and a finite number of measurements is shown below. Given nonvanishing $\Gamma_j$, to guarantee that the estimation $\hat{\Delta}_j$ is close to the true value $\Delta_j$ within error $\varepsilon$, we require the maximum time $\mathcal{O}(\gamma_j^{-1}\log(1/\varepsilon))$ and the number of measurements $\tilde{\mathcal{O}}(\varepsilon^{-4}\Gamma_j^{-2}\gamma_j^4)$. The rigorous description and the proof can be found in Methods.

It is worth noting that the energy excitation of a periodic system typically is a collective behaviour and the spectral weights are concentrated around the allowed excitations. This can be seen from Eq. (4) where the momentum selection rule restricts the allowed excitations to those that satisfy this condition. In addition, the coherence condition will impose restrictions on the possible excitations that can be probed by the initial state and the observable; for example, the selection of certain mode **p** in Eq. (5) in cases where single-particle excitations are primarily concerned. This implies that in certain systems, the total number of allowed excitations may not grow exponentially with respect to the system size.

The key element in this protocol is the measurement of observable dynamics $G(t)$, which can be implemented using Trotterised product formulae which are ancilla-free. Here, we consider using an improved Trotter formula developed in ref. [56], which can effectively eliminate the Trotter error without sacrificing the precision advantage or introducing any ancillary qubits. Suppose the Hamiltonian can be decomposed into Pauli bases as $H = \sum_{l=1}^L \alpha_l P_l$ with $P_l$ being the Pauli operator and $\|\alpha\|_1 := \sum_l |\alpha_l|$. The gate complexity for running the time evolution is $O((\|\alpha\|_1 t)^{1+o(1)} L \log(\varepsilon^{-1}))$, and the small overhead in the power depends on the order of the Trotter algorithm. The asymptotic scaling for lattice models with nearest-neighbour interactions is $O((Nt)^{1+o(1)}\varepsilon^{-o(1)})$ with the system size $N$. In the situation where the evolution time is $t = \mathcal{O}(N\gamma_j^{-1})$, the gate complexity scales as $O(N^{2+o(1)}\varepsilon^{-o(1)}\gamma^{-1})$. We will then numerically investigate the performance of the spectroscopic approach with an increasing system size.

A caveat is that some systems may become thermalised up to a certain time scale, for example, nonintegrable systems. In this situation, the initial state information is lost up to a certain time, and the microscopic feature of the system may not be resolvable by long-time evolution. We note that the spectral information is extracted from a relatively shorter time scale, which to some extent avoids this issue. It would be an interesting direction to discuss the appropriate time scale for resolving the target transitions.

## Error effect in transition energy estimation

The preceding sections have discussed the algorithmic error due to finite simulation time and the uncertainty error. In this section, we will first discuss the algorithmic errors from Trotterisation or imperfect control of the Hamiltonian dynamics. Then, we will discuss errors due to imperfect quantum operations.

The effect of algorithmic errors from Trotterisation or imperfections of the Hamiltonian can be regarded as an additional term $\delta H$ to the ideal Hamiltonian. For lattice models, the Trotter error conserves translation invariance and simply results in a shift to the energy spectrum. The new Hamiltonian also conserves translation invariance $\hat{\mathbf{P}}|\nu\rangle = \mathbf{p}_\nu|\nu\rangle$ where the eigenbasis of the new Hamiltonian is denoted as $|\nu\rangle$. The observable expectation is given by $\langle\nu|\hat{O}(\mathbf{x})|\nu'\rangle = e^{i(\mathbf{p}_{\nu'}-\mathbf{p}_\nu)\mathbf{x}}\langle\nu|\hat{O}|\nu'\rangle$. The momentum selection rule still holds, which imposes $\mathbf{k} = \mathbf{p}_{\nu'} - \mathbf{p}_\nu$ with the excitation between $|\nu'\rangle$ and $|\nu\rangle$ being connected by wavevector $\mathbf{k}$. However, noise will result in a deviation in transition energies. Up to the first-order perturbative expansion, $G(\omega)$ becomes

$$G(\omega) = \sum_{\nu',\nu} \Gamma_{\nu',\nu} p(\tau(\Delta_{\nu',\nu} - \omega)), \tag{7}$$

where $\Delta_{\nu',\nu} = \Delta_{n',n} + \langle n'|\delta H|n'\rangle - \langle n|\delta H|n\rangle$. The error results in a deviation of the resolved energy difference from the ideal one.

The first-order change in the $\nu$th eigenstate is related to the unperturbed one by $|\nu\rangle = |n\rangle + \sum_{m\neq n} A_{mn}|m\rangle$ with $A_{mn} = \Delta_{n,m}^{-1}\langle m|\delta H|n\rangle$. The coherence has a deviation from the unperturbed case,

$$\delta\Gamma_{\nu',\nu} = \alpha_{n'n}\langle n|\hat{O}|n'\rangle + \beta_{n'n}\rho^{n'n}, \tag{8}$$

where $\delta\Gamma_{\nu',\nu} = \Gamma_{\nu',\nu} - \Gamma_{n',n}$, and the coefficients are defined as $\alpha_{n'n} := \sum_m (A_{mn}\rho^{n',m} + A_{mn}^*\rho^{m,n})$ and $\beta_{n'n} := \sum_m (A_{mn'}\langle n|\hat{O}|m\rangle + A_{mn}^*\langle m|\hat{O}|n'\rangle)$ which are related to the initial state and the observable, respectively. Compared to the unperturbed case, some eigenstates $|m\rangle$, which are absent in the original selection rule, also contribute to $\Gamma_{n',n}$ and hence change the spectral weight that can be observed by $G(\omega)$. In the presence of the disordered term which breaks the translation variance, the peaks of $G(\omega)$ will be broadened.

## Resource requirement in the presence of device noise

Finally, we analyse the effect of device noise, its mitigation and the resource requirement due to noise. We start with the discussion on errors in the process of implementation $e^{-iHt}$. Since this is a continuous process, a simple way to describe a physical noisy process is the following: the state remains unaffected with probability $\lambda\delta t$ while becomes a mixed state with probability $1 - \lambda\delta t$, with noise strength characterised by $\lambda$. This process is described by $\mathcal{E}_{\delta t}(\rho) = (1 - \lambda\delta t)\rho + \lambda\delta t\rho_{\text{mix}}$, where $\rho_{\text{mix}} = \frac{I}{2^N}$ is the maximally mixed state and $I$ is the identity matrix. It is easy to see that the noise channel coupled with the unitary operator is only dependent on $t$ and the effective action of such a noise channel is a global depolarising noise. To have an estimate of $\lambda$ under the global noise model, we can fit the experimental results in a similar vein to randomised benchmarking, which could be robust against state preparation and measurement noise[57]. The expectation value of a Pauli operator is

$$\mathbb{E}\hat{o}_{noisy}(t) = \text{Tr}\left(\prod_{\delta t}^{T}(\mathcal{E}_{\delta t}\circ\mathcal{U}_{\delta t})(\rho_0)\right) = \Lambda(t)^{-1}\mathbb{E}\hat{o}_{ideal}(t) \tag{9}$$

with $\Lambda(t) = e^{\lambda t}$. This result indicates that under depolarising types of noise, the ideal expectation value can be obtained by multiplying the factor $\Lambda(t)$ to the noisy result. In some circumstances, the realistic circuit noise may be converted to a global white noise by Pauli twirling[57,58] and as studied in refs. 59,60, shallow quantum circuits can scramble the noise into global white noise. We numerically verify that the global white noise ansatz could be used to approximate physical noise in certain circumstances, such as local depolarising noise, with simulation results shown in Methods.

For more general noise and its mitigation, when the total noise strength is bounded, we can still mitigate errors by probabilistic error cancellation[61–63], and the strategy is discussed in Supplementary Section II.1. Due to error mitigation, the variance of $G(\omega)$ will be amplified by $\max_t\Lambda(t)$. In the presence of noise, the number of samples required by our method maintains polynomial in inverse precision $\mathcal{O}(\text{poly}(1/\epsilon))$. On the other hand, other methods based on the Fourier transform of the time signals or quantum phase estimation will require an amplified number of samples $\mathcal{O}(\exp(1/\epsilon))$. Our method thus shows certain advantages over the existing methods when considering device noise. The simulation results incorporating noise and noise mitigation are shown in Methods. More detailed discussions on error mitigation and resource requirements can be found in Supplementary Section II.

## Numerical studies and experimental demonstration on IBM quantum devices

We investigate the performance of our method for probing the energy excitation spectrum of quantum many-body systems by numerical simulation and experimental demonstration on IBMQ devices. We consider a local perturbation applied to the initial product state, which generates a branch of low-lying excitations of the system. Then the system is evolved under unitaries $U = e^{-iHt_i}$ with different times $t_i$ drawn from the Gaussian distribution $g(t)$, followed by quantum measurement.

We first demonstrate how transition energies can be estimated. Let us consider a one-dimensional (1D) Heisenberg model with an external field

$$H = J\sum_i (\hat{\sigma}_i^x\hat{\sigma}_{i+1}^x + \hat{\sigma}_i^y\hat{\sigma}_{i+1}^y + \hat{\sigma}_i^z\hat{\sigma}_{i+1}^z) + h_z\sum_i \hat{\sigma}_i^z, \tag{10}$$

with the periodic boundary condition, where $\hat{\sigma}_i^\alpha$ is the Pauli operator along the $\alpha$ axis acting on the site $i$, $J = -1$ is the ferromagnetic coupling strength, and $h_z$ is the external magnetic field. To break the ground state degeneracy and make more excitations emerge, a small field $h_z = 10^{-2}J$ aligned in the $z$-direction is applied. The observable is chosen as $\hat{\sigma}_i^y$ on each site, which reverses the quench operation. The transition energy spectrum of the Heisenberg model is shown in Fig. 1a. The peaks become sharper and thus distinguishable with increasing time. We find that the transition energies for the Heisenberg model are estimated very accurately because the dominant transition is distinguishable from the other ones. Figure 1 inset shows the full width of the peak at half maximum (FWHM) with increasing time.

We then show results for estimating transition energies for molecular systems, taking LiH as an example. The molecular Hamiltonian of LiH at the bond length $r = 5.0\text{Å}$ is encoded into six qubits by Jordan-Wigner mapping. The observable is chosen as a particle-conserving operator $\hat{c}_0^\dagger\hat{c}_1 + \hat{c}_0\hat{c}_1^\dagger$ with fermionic (creation) annihilation operator $\hat{c}$ ($\hat{c}^\dagger$), which considers the transitions between low-lying excited states, in order to make $\langle n|\hat{O}|n'\rangle$ non-vanishing. The transition states are marked alongside the corresponding energy resonance in Fig. 1b. In molecular systems, the excited states of molecular systems are often closely spaced, resulting in interference between different transitions. As a result, the visible peak appears as an addition of different peaks, making it difficult to locate the true transitions from the peak. However, as time increases, the peaks become sharper and more peaks appear, which allows for the distinction of the true transition. The inset of Fig. 1b shows the estimation error with increasing time for the dominant transition. It is worth noting the initial state remains a low-lying excitation above the ground state, and the first few low-lying excited states are degenerate. As molecules become more complex, it is anticipated that more near-degenerate low-lying states will emerge,

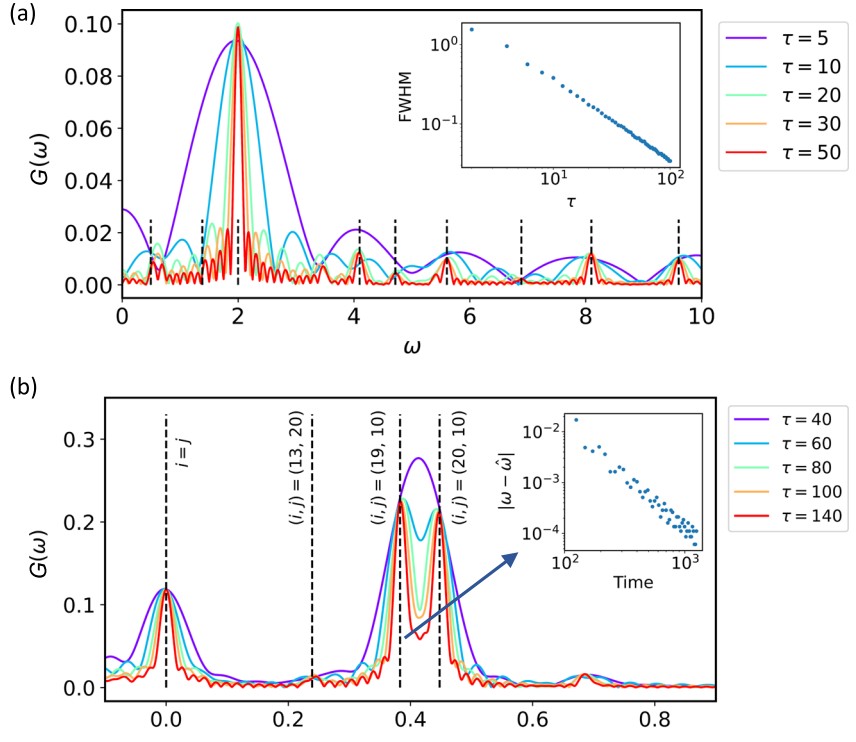

**Fig. 1 | Transition energy spectra search of the Heisenberg model and the LiH molecule. a** The 7-site Heisenberg model. The state is first initialised in a product state $|+\rangle^{\otimes N}$, and then a rotation is applied to the central qubit, which takes the form of $\hat{\sigma}_3^x |+\rangle^{\otimes N}$, where the qubit numbering starts from zero in this work. The dashed vertical lines show the ideal transition energies whose coherence $\Gamma_{i,j}$ is above a certain threshold, which are calculated by exact diagonalisation (ED). Inset: FWHM dependence on varying $\tau$. The cutoff is chosen as $T_{cut} = 1$. **b** The excitation spectrum for LiH. The initial state is prepared as a Hartree-Fock state. The vertical lines show the ideal transition energies $\Delta_{i,j}$ calculated by ED. The transition state is marked alongside the corresponding energy resonance. Inset: Estimation error for the dominant transition energy indicated by the arrow with varying total time.

leading to an increase in peak interference and hence a more intricate spectrum. To distinguish small, adjacent peaks, we need to increase the evolution time. The results in Fig. 1 show how transition energies can be estimated and how the simulation accuracy can be improved by increasing the simulation time. The evolution of peaks of other molecules with varying time and their spectral weights can be found in Supplementary Section IV.

Next, we show how the excitation spectra of lattice models can be probed with the IBM Kolkata quantum device. We consider the 1D Ising model $H = \sum_i \hat{\sigma}_i^z \hat{\sigma}_{i+1}^z + 2\sum_i \hat{\sigma}_i^x$ with nearest-neighbour interaction. The ground state is close to the product state when the external field is large, and thus we consider the excitation being generated by a local perturbation. The initial state is prepared as $|\psi_0\rangle = R_y^i(\frac{\pi}{2})|+\rangle^{\otimes N}$ with the single-qubit rotation operation acting on the central site $i$. Similarly, we demonstrate the simulation of the 1D Heisenberg model in Eq. (10) with $h_z = 0$ and $J = 1$. The time evolution is simulated using Trotter formulae with an even-odd order pairing method such that the Trotter error is reduced. More details about the circuit compilation can be found in Methods. The excitation spectra of the Ising model and the Heisenberg model are shown in Fig. 2a, b, respectively. The energy dispersions for both models are in good agreement with the analytic results for infinitely long spin chains. The experimental simulation errors for the observable dynamics and energy excitations are shown in Fig. 2d, e, respectively. The simulation on the Kolkata device involves over 350 CNOT gates, but the simulation error is maintained at an acceptable level, which indicates the robustness of our method.

The maximum system size is restricted by the large noise and the available size of the hardware. To further show the performance of our

method for a relatively large system, we numerically test our method with an increasing system size using tensor network methods. We show the simulation error with different system sizes up to 51 qubits in Fig. 2f. The error is decreased by increasing $\tau$, and it is not increased with increasing system size. In Fig. 2c, we show the excitation spectra of the 51-site Heisenberg model for comparison with the experimental result in Fig. 2b. We refer to Supplementary Section IV for more results of excitation spectra simulation of the 2D Heisenberg model, the Bose-Hubbard and the Fermi-Hubbard model. Note that the examples studied here are classically simulable by tensor networks since the central aim is to test the performance of our method for known systems under different conditions. We do not attempt to prove a quantum advantage over classical computing or show its generality for solving arbitrary quantum systems. Instead, we aim to show how the spectroscopic properties of many-body systems can be probed using either analogue or digital quantum simulators as a proof of concept.

We remark that simulating general gapless systems is a widely believed challenge, which poses great challenges to most existing algorithms, such as phase estimation and quantum singular value transformation[64]. The difficulty is that the energies are highly degenerate such that energy is not sufficient to distinguish these states. By introducing other conjugate variables, we could label the states with both the energy and the auxiliary characters (e.g. energy and momentum $(E, k)$) such that the degenerated states can be distinguished. Therefore, this strategy may be useful for analysing gapless systems with certain symmetry conservation, like translation invariance. Although we only test for simple and specific examples in this work, the results might shed light on the resolution of energy spectra of other gapless systems. The spectroscopic method and the examples

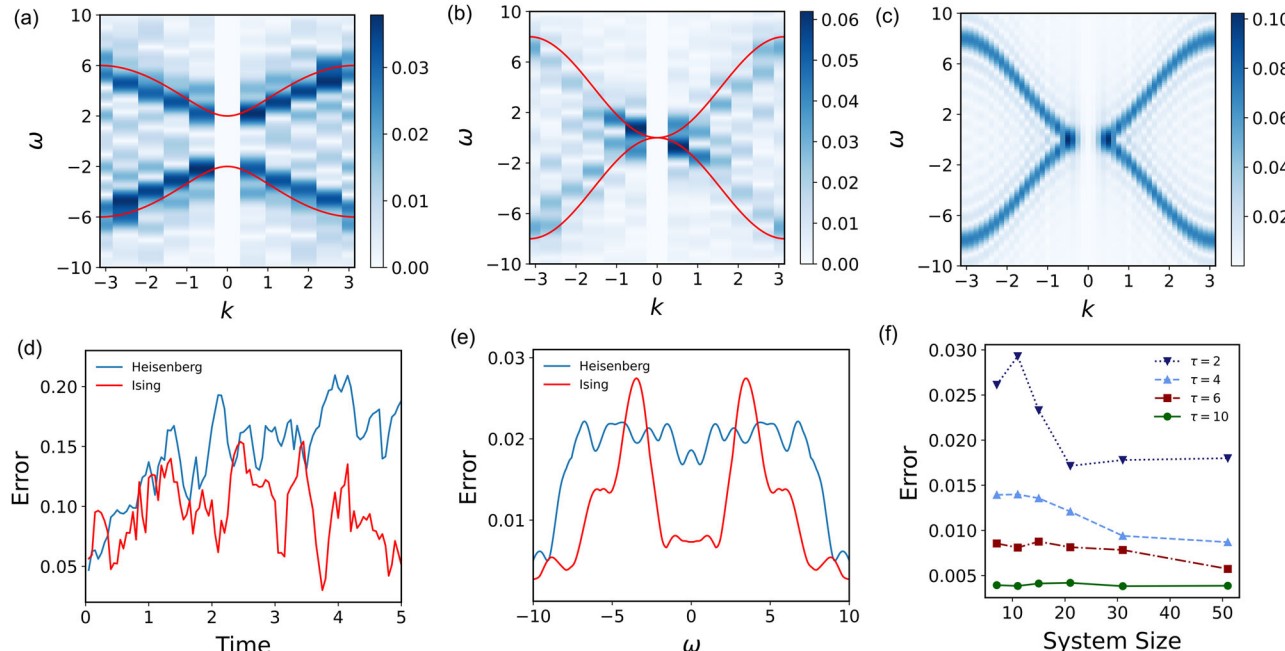

**Fig. 2 | The energy dispersions of 1D lattice models through the demonstration on the IBM quantum devices and tensor-network simulation. a** The excitation spectra $G(\omega)$ of an 11-site Ising model. **b** The excitation spectra of a 13-site Heisenberg model. The results are obtained from measurements on IBMQ devices. The red lines in **a** and **b** represent the energy dispersions of infinite long spin chains obtained by analytic calculations. The intensities at $\mathbf{k} = 0$ have been removed. **c** The energy dispersion for the 1D Heisenberg chain of length $N = 51$ with $h_z = 0$ and $J = 1$. The total time evolution is set as $T_{tot} = \tau = 5$. **d** Experimental simulation error for the measurement results of $\sum_i \langle \hat{\sigma}_i^y(t) \rangle / N$ under real-time evolution conducted on IBMQ devices, which is compared to results using noiseless Trotter formulae for

both the Ising Hamiltonian (red line) and the Heisenberg Hamiltonian (blue line). **e** Experimental simulation error for the measurement results $G(\omega)$ from IBMQ experiments in the frequency domain averaged over different momenta. The results in **d** and **e** are compared to those using noiseless Trotter formulae. The maximal time length is $T_{tot} = 5$. **f** Scalability analysis for the gapless Heisenberg chain. The numerical simulation results for different values of $\tau$ are compared with the approximately ideal case in which $\tau \to \infty$. The figure shows the maximum error of the intensity $G(\omega)$ in the momentum space for different time scales $\tau$ and system sizes. The error is calculated as $\max_k |\overline{G_{k,\tau}(\omega) - G_{k,\tau\to\infty}(\omega)}|$ where the averaging has been taken over $\omega$.

examined here could potentially stimulate further discussions on estimating the properties of gapless systems.

## Discussion

In this work, we introduce a spectroscopic method for probing transition energies and excitation spectra of quantum many-body systems. The framework presented in this work establishes a connection between spectroscopic techniques and quantum simulation, and thus enables mutual benefits and advancements in both fields. The key element is the realisation of frequency resonance, and one can detect systems with certain symmetries by introducing additional momentum resonance. Our method requires the implementation of short-time dynamics only with the time length logarithmic in precision, and could be experiment-friendly for the current generation of quantum simulators. Advanced measurement algorithms[65,66] (e.g. classical shadows[35]) can be directly employed to reduce the measurement complexity in measuring observables within this framework (see Methods for details).

We numerically test the scalability of the spectroscopic method for lattice models including both gapped and gapless cases. The simulation results obtained for gapless systems with translation invariance imply that our method may be useful for analysing similar systems with certain symmetry conservation, and we leave it to future work. We note that when the ground-state energy is known, finding the transition energy between the excited state $|n\rangle$ and the ground state $|0\rangle$ is equivalent to the problem of finding the eigenenergy of a quantum system. Therefore, resolving the transition energy could be quantumly hard. Our method is efficient when the state and observable dependent coherence is nonvanishing which could be satisfied in several quantum systems with the conservation of particle numbers.

Though this in general does not hold, we tested the coherence for molecules and lattice models. On the other hand, we remark that although we do not expect that our method can resolve the transition energy for general systems, the spectroscopic protocol can find applications in simulating the energy excitation spectrum; in this task the excitation is a collective behaviour and hence we do not necessarily require resolving a specific energy transition. In addition, the simulated spectroscopic features could be useful for identifying the transition from a Fermi-liquid to a non-Fermi-liquid phase and pinpointing the breakdown of where the quasiparticle picture no longer holds.

The simulation result suggests that the spectral properties of several quantum many-body systems can be estimated with high accuracy, even with increasing system sizes for lattice Hamiltonians considered in this work and when statistical and device noise are present. We also demonstrated how our method can be applied to probe the excitation spectra of spin Hamiltonians on IBM quantum devices. These results indicate that our approach could serve as a quantum computing solution complementary to high-intensity scattering facilities. For example, it could be useful in stimulating new quantum computing-based methods for resolving complex many-body systems with different conditions, such as various materials-dependent conditions (e.g. doping levels) and environmental conditions (e.g. pressure and temperature).

## Methods
### Equivalence of the two formalisms
The function $p(\cdot)$ plays a role as a spectral detector that can filter $\Delta_{n'n}$ out of the other transition energies. In this work, we focus on the Gaussian function since the truncation error is small. Other functions

can be chosen, and the resource requirement can be easily analysed under our framework.

If $\tau$ is introduced as a separate parameter that is irrelevant to the Fourier transform, it is straightforward to have $\tilde{g}_\tau(t) := \int_{-\infty}^{+\infty} p_\tau(\omega) e^{i\omega t} d\omega$ and $p_\tau(\omega) = \int_{-\infty}^{+\infty} \tilde{g}_\tau(t) e^{-i\omega t} dt$. One can check that $\tilde{g}_\tau(t) = \frac{1}{\tau}\tilde{g}(\frac{t}{\tau})$, where $\tilde{g}$ was defined in the main text. We have

$$p_\tau(\omega) = \frac{c(\tau)}{2\pi} \int_{-\infty}^{+\infty} \Pr(t,\tau) e^{i\theta_t} e^{-i\omega t} dt \tag{11}$$

where $c(\tau) := \int_{-\infty}^{+\infty} |\tilde{g}_\tau(t)| dt = c$ and $\Pr(t,\tau) := |\tilde{g}_\tau(t)|/c(\tau)$. On the other hand, we can arrange that $\tau$ is coupled with $\omega$ in the Fourier transform, which is the way introduced in the main text. One can check that Eq. (11) is equivalent to $p(\tau\omega)$ defined in the main text, and thus the two formalisms are equivalent. One important point to highlight is that the function $g(t)$ is $\tau$-independent and is normalised. That means we can easily analyse and compare different methods in a unified framework, rather than a case-by-case analysis. Therefore, the method presented in the main text facilitates a straightforward evaluation of the resource requirement. More detailed derivations can be found in Supplementary Section I.

We remark that our approach is capable of estimating the energy differences between energy eigenstates by identifying the peaks in the excitation spectrum. However, applying the spectral filter will alter the width of the peak, in contrast to conventional spectroscopic techniques. Therefore, properties in relation to the peak width, such as the lifetime of quasiparticles, cannot be determined. In addition, the introduction of a filter function will alter the width of the peak. Therefore, we cannot distinguish whether the broadening of the peak is caused by a finite evolution time or a continuum, because both can lead to a broadening of the peak.

## Analysis of the algorithmic error and resource requirements

In this section, we first discuss the coherence in particle-conserved systems. Then, we analyse the estimation error of the transition energies under a finite imaginary time $\tau$, a finite cutoff $T$ when evaluating the integral, and a finite number of measurements $N_s$. Based on the error analysis, we estimate the resource requirement for transition energy estimation.

The observable is assumed to have a bounded norm $\| \hat{O} \| \leq 1$; for instance, the observable can be chosen as a tensor product of single-qubit Pauli operators. Consequently, we have $|\mathrm{Tr}(\hat{O}\rho)| \leq 1$ and $\max_j \Gamma_j \leq 1$. As $\mathrm{Tr}(\hat{O}\rho) = \sum_{n,n'} \rho^{n'n} \langle n|\hat{O}|n'\rangle$, the sum of the spectral weight has an upper bound $\sum_{n,n'} \Gamma_{n'n} \leq 1$. We will discuss the condition for ensuring a nonzero coherence in particle-conserved systems in the following.

In the main text, we showed that for certain many-body systems, by measuring the (quasi)particle annihilation and creation operator $\hat{O} = \sum_{\mathbf{p}'} A_{\mathbf{p}'}\hat{\gamma}_{\mathbf{p}'}^\dagger + A_{\mathbf{p}'}^*\hat{\gamma}_{\mathbf{p}'}$, we can observe the transitions (since $\Gamma_{n,0} \neq 0$). In a special case, for Bose-Hubbard models, as has been demonstrated experimentally for the hard-core boson cases in ref. 22, we may consider the measurement of $\hat{a}_r$ which is the original bosonic annihilation operator on the $r$th site in this context. This is a special case of our choice. Since the $\hat{a}_r = \hat{x}_r + i\hat{p}_r$ where $\hat{x}_r$ and $\hat{p}_r$ are the canonical coordinates for the $r$th mode, the measurement of $\langle \hat{a}_r \rangle$ could be realised by measuring in the single-mode canonical coordinates and then post-processing the measurement outcomes.

To probe $\Gamma_{n',n}$, we could choose to initialise the state by creating two particles above the vacuum state as $|\psi_0\rangle = \frac{1}{1+\beta^2}(1+\beta\hat{\gamma}_{\mathbf{p}}^\dagger)(1+\beta\hat{\gamma}_{\mathbf{p}'}^\dagger)|0\rangle$. Again, we can decompose the single-particle state $\hat{\gamma}_{\mathbf{p}}^\dagger|0\rangle$ with momentum $\mathbf{p}$ into the single-particle eigenbasis of $|n\rangle$, and we denote the expansion coefficient $\langle n|\hat{\gamma}_{\mathbf{p}}^\dagger|0\rangle := c_{n,\mathbf{p}}$ with the normalisation condition $\sum_n |c_{n,\mathbf{p}}|^2 = 1$. We could find that in the case where the number of possible excitations grows polynomially with respect to the system size the coefficient will be nonvanishing. Similarly, for the

two-particle state, we can decompose it into the two-particle eigenbasis of $|n^{(2)}\rangle$, and we denote the expansion coefficient $\langle n^{(2)}|\hat{\gamma}_{\mathbf{p}}^\dagger\hat{\gamma}_{\mathbf{p}'}^\dagger|0\rangle := c_{n,\mathbf{pp}'}^{(2)}$.

If we simply wish to observe the transitions in the single-particle manifolds, then for the observable $\hat{O} = \sum_{\mathbf{q},\mathbf{q}'} A_{\mathbf{q},\mathbf{q}'}\hat{\gamma}_{\mathbf{q}}^\dagger\hat{\gamma}_{\mathbf{q}'}$, the transition amplitude between single-particle eigenstate $|n\rangle$ and $|n'\rangle$ may be approximated by $\langle n|\hat{O}|n'\rangle = \sum_{\mathbf{q},\mathbf{q}'} A_{\mathbf{q},\mathbf{q}'}c_{n,\mathbf{q}}c_{\mathbf{q}',n'}^*$. Here we have put a strong assumption that the space of eigenstates with different particle numbers is orthogonal, specifically $\langle n|\hat{\gamma}_{\mathbf{q}}^\dagger|m\rangle = 0$ for $m \neq 0$. We conjecture that this condition holds when the quasiparticle number is well-defined. The transitions in the many-particle manifolds are more complicated; we may deliberately choose the observable to infer the transition energy. In our case, the initial state coherence is

$$\rho^{n',n} = \frac{\beta^2}{(1+\beta^2)^2}(c_{n',\mathbf{p}'} + c_{n',\mathbf{p}})(c_{\mathbf{p}',n}^* + c_{\mathbf{p},n}^*).$$

To sum up, the coherence is given by

$$\Gamma_{n',n} = \sum_{\mathbf{q},\mathbf{q}'} \frac{\beta^2 A_{\mathbf{q},\mathbf{q}'}c_{n,\mathbf{q}}c_{\mathbf{q}',n'}^*}{(1+\beta^2)^2}(c_{n',\mathbf{p}'} + c_{n',\mathbf{p}})(c_{\mathbf{p}',n}^* + c_{\mathbf{p},n}^*). \tag{12}$$

We give a few comments on the coherence in many-body quantum systems. As discussed in the main text, systems that host quasiparticles satisfy the condition for a nonvanishing coherence. For Fermi liquid, the low-energy eigenstates are labelled by a set of quantum numbers $n_{k,\sigma} = 0, 1$ (the occupation numbers) and the introduction of interactions will not let the quasiparticle decay into multiple other quasiparticles[50]. The one quasiparticle state has a finite overlap with the state where a bare particle carrying the same quantum numbers is added. More concretely, the overlap between the quasiparticle state $|\psi_k\rangle$ carrying momentum $k$ and the ground state $|\psi_0\rangle$ excited by $\hat{c}_{k,\sigma}^\dagger$, $|\langle\psi_k|\hat{c}_{k,\sigma}^\dagger|\psi_0\rangle|^2$, is not zero in the thermodynamic limit for $k$ near the Fermi surface. Note that the overlap is between 0 and 1 because the quasiparticle state $|\psi_k\rangle$ is a superposition of all states with momentum $k$. This property gives a guarantee for the nonzero coherence. The non-metallic states such as anti-ferromagnetic states are generated due to interactions between electrons and can be understood from this Fermi liquid theory[50]. Nevertheless, it is worth noting that finding the occupancy of the original orbitals in the interacting eigenstate $|n'\rangle$, $n_{\mathrm{occ}}(k,\sigma) = \langle n'|\hat{c}_{k,\sigma}^\dagger\hat{c}_{k,\sigma}|n'\rangle$ is still a computationally hard task. The simulated spectroscopic features could thus be useful for understanding certain behaviours of many-body systems.

Although our method relies on the coherence between the target eigenstates, it could be useful in identifying certain spectroscopic features of the target system. For example, in the case of non-Fermi liquid, the lifetime of quasiparticles is short and its spectrum is expected to be blurred. This blurred spectroscopic feature indicates the beginning of non-Fermi-liquid behaviour. Understanding this blurred behaviour in nature is an interesting question, and our approach can help reveal it. The simulated spectroscopic features could be useful for identifying the transition from a Fermi liquid to a non-Fermi-liquid phase and pinpointing the breakdown of where Fermi liquid theory no longer holds.

Below, we will assume $\Gamma_{n',n}$ is nonzero and give the circuit complexity dependence on $\Gamma_{n',n}$. Recall that the spectral detector takes the form of

$$G(\omega) := \sum_{n,n' \geq 0} \Gamma_{n',n} p(\tau(E_{n'} - E_n - \omega)), \tag{13}$$

which can be simplified as $G(\omega) = 2\sum_{n<n'} \Re(\Gamma_{n',n}) p(\tau(E_{n'} - E_n - \omega))$. For simplicity, we sort $\Delta_{n',n}$ in ascending order and denote ordered energies as $\Delta_i$ and the associated coherence is denoted as $\Gamma_i \leftrightarrow 2\Re(\Gamma_{n',n}) \in \mathbb{R}$. The spectral detector can now be expressed as

$G(\omega) = \sum_i \Gamma_i p(\tau(\Delta_i - \omega))$. Then, the original problem is converted to a standard energy estimation problem[37,39].

Suppose we aim to estimate $\Delta_j$ with a nonvanishing $\Gamma_j$. Without loss of generality, we assume $\Gamma_j > 0$ in the following discussion, such that $G(\omega)$ is positive when $\omega$ approaches $\Delta_j$ (given a large $\Gamma_j$). Similar results can be obtained in the case of negative $\Gamma_j$. Intuitively, the spectral information can be inferred by looking at the peak of the intensity of $|G(\omega)|$. This is guaranteed by the following lemma.

**Lemma 1.** When $\tau = \frac{1}{0.9\gamma}\sqrt{\ln\left(\frac{20}{\varepsilon^2\Gamma_j}\right)}$ and $\varepsilon \leq 0.2\gamma$, the two inequalities hold:

$$\Gamma_j - G(\omega) \leq 0.3\tau^2\varepsilon^2\Gamma_j, \forall|\omega - \Delta_j| \leq 0.5\varepsilon \qquad (14)$$

and

$$\Gamma_j - G(\omega) \geq 0.8\tau^2\varepsilon^2\Gamma_j, \forall|\omega - \Delta_j| \in (\varepsilon, 0.1\gamma) \qquad (15)$$

The above lemma guarantees that the distance $d = |\Gamma_j - \hat{G}(\omega)|$ is modulated by the estimation error $|\omega - \Delta_j|$. Therefore, $\Delta_j$ can be distinguished from the other transitions when $\omega$ approaches $\Delta_j$. The next step is to analyse the quantum resources required to have a good estimation of $\hat{G}(\omega)$.

When evaluating the integral in Eq. (3) in the main text, we set a finite cutoff for the integral range from $[\infty, \infty]$ to $[-T, T]$ in practice. The truncated detector of $\hat{G}(\omega)$ is defined as

$$G^{(T)}(\omega) = \int_T^T G(\tau t)g(t)e^{i\theta_t}e^{it\omega t}dt. \qquad (16)$$

If a Gaussian operation is used, it is easy to check that the truncation error has an upper bound

$$|G^{(T)}(\omega) - G(\omega)| \leq 2\int_T^\infty g(t)dt = \text{erfc}(T/2) \leq \exp(-T^2/4) \qquad (17)$$

which has been discussed in prior work[37,39]. Therefore, the truncation error can be bounded by

$$|G^{(T)}(\omega) - G(\omega)| \leq \varepsilon_T, \forall\omega \in \mathbb{R} \qquad (18)$$

when the cutoff time $T \geq 2\sqrt{\ln(1/\varepsilon_T)}$.

In the following, we consider the error due to a finite number of measurements. A single-shot estimator of $G^{(T)}(\omega)$ takes the form of

$$\hat{G}_i^{(T)}(\omega) = \begin{cases} \hat{o}(\tau t_i)e^{i\theta_{t_i}}e^{it\omega t_i}, & |t_i| \leq T \\ 0, & |t_i| > T \end{cases} \qquad (19)$$

where the time length $t_i$ is drawn from the probability distribution $g(t)$, which is $\tau$-independent, and $\hat{o}(\tau t_i)$ is an unbiased estimate of $\text{Tr}[\hat{O}(\tau t)\rho]$. We can estimate $G(\omega)$ by

$$\hat{G}(\omega) = \frac{1}{N_s}\sum_{i=1}^{N_s}\hat{G}_i^{(T)}(\omega), \qquad (20)$$

where $N_s$ is the total number of samples. It is worth noting at this juncture that both $\text{Tr}[\hat{O}(\tau t)\rho]$ and $G(\omega)$ are real numbers, in contrast to the cases in eigenenergy estimation where the expectation values are complex numbers.

The transition energy is estimated by

$$\hat{\Delta}_j = \text{argmax}_{\omega \in [\Delta_j - \gamma/2, \Delta_j + \gamma/2]}\hat{G}^{(T)}(\omega). \qquad (21)$$

Note that it is assumed that $\Gamma_j > 0$. The result is summarised as follows.

**Proposition 1.** To guarantee that an estimation $\hat{\Delta}_j$ of the transition energy $\Delta_j$ is close to the true value within an error $\varepsilon$, with a failure probability $\delta$, we require that the maximum time is $\mathcal{O}(\gamma^{-1}\log(1/\varepsilon))$ and the number of measurements $\tilde{\mathcal{O}}(\varepsilon^{-4}\Gamma_j^{-2}\gamma^3\log(1/\delta))$, where $\gamma$ is a chosen lower bound of the gap difference $\gamma \leq \min\{\Delta_{j+1} - \Delta_j, \Delta_j - \Delta_{j-1}\}$ where $j$ runs over the allowed excitations restricted by the energy and momentum selection rule.

In this work we use the $\tilde{\mathcal{O}}$ notation where the polylogarithmic dependence is hidden. Proposition 1 indicates that the maximum circuit depth is logarithmic in precision and the total running time scales as $\tilde{\mathcal{O}}(\varepsilon^{-4})$, in contrast to the result established in ref. 37 or ref. 30 which is $\tilde{\mathcal{O}}(\varepsilon^{-1})$. The above result is guaranteed by Lemma 2 in Supplementary Information, which shows that the maximum time required is $t = \tau \times T = \mathcal{O}(\gamma^{-1}\log(\varepsilon^{-1}))$ and $N_s = \tilde{\mathcal{O}}(\varepsilon^{-4}\gamma^4\Gamma_j^{-2}\log(\delta^{-1}))$. The total running time scales as $\tau \times T \times N_s = \tilde{\mathcal{O}}(\varepsilon^{-4}\gamma^3\log(\varepsilon^{-1})\log(\delta^{-1}))$. Proposition 1 naturally flows from the above analysis. We leave the proof for Lemma 1 and Lemma 2 to Supplementary Section I.

## Complexity for measuring multiple observables

In the above analysis, we assume that $\|\hat{O}\| \leq 1$. It is easy to find that our method only requires measurement of $\langle\psi(t)|\hat{O}|\psi(t)\rangle$ and thus any kind of measurement scheme is directly applicable. Therefore, multiple low-support observables can be estimated in an efficient way. Suppose there are $N_o$ low-support observables to be measured and the conditions in Proposition 1 are satisfied. By employing classical shadow methods developed in ref. 35, the total time complexity is $\mathcal{O}(\gamma^3\varepsilon^{-4}\log(1/\varepsilon)\log(N_o))$, which is only amplified by a logarithmic factor compared to single observable estimation. Note that the measurement complexity scales exponentially with respect to the locality of the observables $m$ as $\mathcal{O}(3^m)$. Advanced measurement algorithms can be directly employed within our framework, such as Pauli grouping methods[65,66], whose efficiency in measuring qubit-wise compatible observables has been verified for various examples.

Recently Chan et al. proposed to extract eigenenergy differences by post-processing classical shadows of time-evolved states[25]. The quantum resources and the signal-to-noise ratio were analysed. Since the extraction of eigenenergy differences is based on the post-processing of many time-evolved states, the low measurement cost is essential for their method. In this work, we show that for several quantum many-body systems, the observables are chosen with clear physical intuitions, which, to some extent, avoids the necessity of measuring many observables.

## Numerical simulation based on tensor networks

For the numerical simulation conducted in this work, the time evolution $e^{-iHt}$ is simulated using a time-dependent variational principle (TDVP) algorithm based on the representation by matrix product states (MPS). The density matrix renormalisation group (DMRG) method is used to initialise the system into the ground state of the system before applying quench. The following shows a few examples of excitation spectra simulation. In Supplementary Section IV, we show simulations of both the Bose-Hubbard and Fermi-Hubbard models. The Hamiltonian for both bosons (BH) and fermions (FH) can be expressed as

$$H_{\text{BH}} = -J\sum_i\left(\hat{a}_i^\dagger\hat{a}_{i+1} + \text{h.c}\right) + \frac{U}{2}\sum_i\hat{n}_i(\hat{n}_i - 1),$$

$$H_{\text{FH}} = -t\sum_i\left(\hat{c}_i^\dagger\hat{c}_{i+1} + \text{h.c}\right) + U\sum_i\hat{n}_{i\uparrow}\hat{n}_{i\downarrow},$$

where $\hat{a}_i$ and $\hat{a}_i^\dagger$ ($\hat{c}_i$ and $\hat{c}_i^\dagger$) are the bosonic (fermionic) annihilation and creation operators at the site $i$, respectively, and $\hat{n}_i$ is the number operator acting on site $i$. In what follows, we examine whether our method does indeed replicate the true excitation spectra. For these

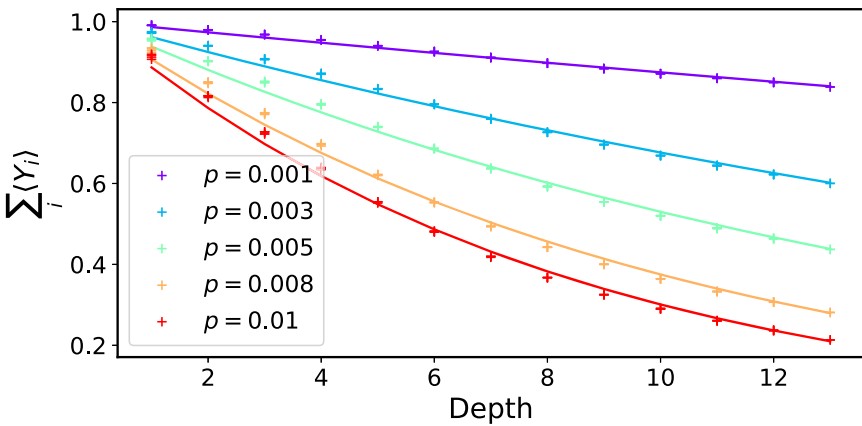

**Fig. 3 | Fitting for the circuit evolution with local depolarising noise.** Numerical results of the circuit $UU^\dagger$ implementation of noisy spectroscopy protocol for simulating the 7-site Ising model ($h_x = 2$ and $h_z = 0.1$) with different noise strength $p$. The depth refers to the repetition numbers. The solid lines represent the exponential fit for every $p$. All the regressions have a relative predictive power $R^2 = 0.99$.

**Fig. 4 | Noisy and error-mitigated time dynamics of the spectroscopy protocol.**
**a** Dynamics of the observable expectation value $\langle Y_i \rangle$ at site $i = 1$. The figure shows the results of the noisy (with noise rate $p = 0.005$), the error mitigated and the ideal cases for a 7-site Ising model. **b** Root mean square error (RMSE) of the noisy and the error mitigated results in the time domain. **c** Error of the noisy and error mitigated spectrum in the frequency domain. The green line represents the maximum value of the error in the time domain as a reference. **d** Error in the $k$-space.

types of particle systems (both bosons and fermions), we consider local perturbation by taking or adding particles to the site that we want to perturb. In this case, we remove all the particles in the central site of the chain. It is easy to see that this operation is not unitary and does not conserve the particle number, though we can find the unitary counterpart of this operation.

For the Hubbard models, we find the ground state using DMRG, apply local perturbation, evolve the system under the Hamiltonian, and finally measure the expectation value of the number operator. The results for the bosonic chain with an average filling of $\bar{n} = 1.4$ and $U/J = 2$ and for the fermionic chain simulation with $h/J = 2$ are shown in Supplementary Fig. 7. In all the 1D simulations, we use bond dimension $\chi = 128$ and chains of up to $N = 51$ sites. For the 2D cases, we need to increase the bond dimension in some cases due to the non-local nature of the MPS when having to simulate an extra dimension. We consider 2D lattices of dimensions $L_x \times L_y$ where $L_x = L_y = 11$. The simulation result for the lattice model with nearest-neighbour interaction is shown in Supplementary Fig. 8. The focus here is to test the effectiveness of the spectroscopic method in a proof-of-principle way.

### Simulation results when considering device and statistical noise

Here, we include device and statistical noise in the simulation. As discussed in the main text, the total running time complexity in the presence of device noise is polynomial $\mathcal{O}(\mathrm{poly}(\epsilon^{-1}))$ in order to achieve a given precision. For global depolarising noise, the noise effect on observable estimation can be analytically derived. Given the knowledge of the error model, we can obtain error-mitigated observable estimation by fitting the noisy results.

Next, we test the performance of the simple error mitigation strategy by considering a more practical setup with a depolarising noise model. Specifically, we introduce local depolarising noise after each gate, including both single- and two-qubit gates. We numerically verify the behaviour of this noise model by emulating the noisy quantum circuits, where we apply local depolarising noise after each gate. We run the circuit for a Trotter step with time interval $\delta t$ followed by its inverse: $U(\delta t)U^\dagger(\delta t)$, which is just identity without noise, and repeat it for different times. In Fig. 3, we show the results of $\sum_i \langle Y_i \rangle$ with an increasing circuit depth. Figure 3 indicates that the noisy results can be well-approximated by an exponential decay function aligning with Eq. (9). The fitting results for individual observable $\langle Y_i \rangle$ are provided in Supplementary Section II.1.

For the noisy simulation and its error mitigation, we consider the 1D Ising model considered in the main text with an additional term $h_z = 0.1$. We set the initial state the same as that in the main text, a maximum time evolution of $T = 5$ and the time interval $\delta t = 0.4$. Results for varying system sizes (up to 11 sites) and noise rates are provided in Supplementary Section II.1. We mitigate the noisy measurement outcomes by using the fitted exponential functions for each qubit. The results under different conditions (noisy, ideal, and error mitigated) are shown in both the time and frequency domains in Fig. 4. We observe that even this simple error mitigation strategy can improve the results, though error mitigation is still needed for more general types of noise. As expected, the algorithm demonstrates strong resilience to noise, allowing us to recover the spectrum even after noisy evolution. This explains why the error is significantly smaller either in the frequency domain or the momentum space.

### Compilation into quantum gates and implementation on IBM quantum devices

After some initial tests on the available quantum devices, the Kolkata device performed consistently for this task and thus was selected for executing the quantum circuits. We selected up to 13 qubits with good readout fidelities and gate fidelities. We used the second-order Trotter formula to simulate the dynamics. The real-time dynamics are compiled into single-qubit Pauli rotation gates and CNOT gates.

For the Heisenberg model, the time-evolution operator $e^{-it\sum_i(X_iX_{i+1} + Y_iY_{i+1} + Z_iZ_{i+1})}$ will be realised by Trotter formulae. Each component $e^{-ix(X_iX_{i+1} + Y_iY_{i+1} + Z_iZ_{i+1})}$ with time duration $x$ can be realised by 3 CNOT gates as proposed by[67]. For instance, $e^{-ix(X_1X_2 + Y_1Y_2 + Z_1Z_2)}$ can be realised by the following circuit,

$$R_z^{(1)}\left(\frac{\pi}{2}\right)\mathrm{CNOT}_{2\to1}R_y^{(2)}\left(\frac{\pi}{2} - 2x\right)\mathrm{CNOT}_{1\to2}$$
$$R_y^{(2)}\left(2x - \frac{\pi}{2}\right)R_z^{(1)}\left(\frac{\pi}{2} - 2x\right)\mathrm{CNOT}_{2\to1}R_z^{(2)}\left(-\frac{\pi}{2}\right)$$

This saves more gates when compared to using a naive compilation which needs 6 CNOT gates.

## Data availability

Source data are provided with this paper. Other data that support the findings of this study can be found on GitHub at https://github.com/luciavilchez/probing_spec_features. Source data are provided with this paper.

## Code availability

Codes developed in the current study and the relevant examples can be accessed on GitHub at https://github.com/luciavilchez/probing_spec_features.

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

## Acknowledgements

This work was initiated when J.S. was a postgraduate student at Oxford as part of his DPhil thesis, see Chapter 7 in ref. 68. L.V.E. and J.S. thank Louis Villa and Steven J. Thomson for discussions on tensor network simulation. J.S. thanks Weitang Li, Andrew Green, Pei Zeng, Chonghao Zhai, and Thomas Cheng for related and valuable discussions on this work. We acknowledge the use of the IBM quantum services for this work. The views expressed are those of the authors and do not reflect the official policy or position of IBM or the IBM quantum team. J.S. and M.S.K. acknowledge the Samsung GRC grant for financial support. V.V. thanks support from the Gordon and Betty Moore Foundation. J.S. thanks support from the Innovate UK (Project No.10075020). This work was also supported by EPSRC through EP/T001062/1, the National Research Foundation of Korea (NRF) grant funded by the Korea government (MSIT) (No. RS-2024-00413957) and Schmidt Sciences, LLC. L.V.E is supported by the Clarendon Fund.

## Author contributions

J.S., L.V.E., V.V., and A.B. conceived the idea. J.S. developed the theoretical aspect of the project. L.V.E. performed the numerical simulations with input from J.S.. J.S. performed simulations using IBM cloud services with input from M.S.K. V.V., A.B. and M.S.K. supervised the project. J.S. and L.V.E. wrote the manuscript. All the authors contributed to the discussion and writing up the manuscript.

## Competing interests

The authors declare no competing interests.
