## [Transparent Peer Review file · Nature Communications]

Probing spectral features of quantum many-body systems with quantum simulators

Corresponding Author: Dr Jinzhao Sun

Version 0:

Reviewer comments:

Reviewer #1

(Remarks to the Author)

The manuscript by Sun and co-workers proposed an approach that relies on sampling time intervals based on a given probability distribution and the evaluation of the time evolution operator at these time points using quantum computers. The idea of reframing the integration over time to some statistical averaging is interesting in the context of quantum computing (not so new in the context of classical simulations). Requiring only relatively short real-time evolution via Trotterization does not require ancilla qubits, providing advantages over some other quantum algorithms (but not all). The new approach can be useful for many systems, but not all, given its requirement for non-vanishing coherence. Whether the novelty of this work meets the standard of nature communications is up to the editor, but I have a few comments that I think need to be addressed before publication.

The manuscript compares their methods with some other methods where two-point unequal time correlation functions are needed for spectral simulations. Since these methods may need ancilla qubits, the proposed method has an advantage. However, this is a bit misleading. Some of these methods were intended to compute actual spectra that can be measured experimentally rather than just the energy spectrum (here, experiment means actual spectroscopic measurement, not quantum simulation on actual quantum computers). On the other hand, the current protocol designed a spectral detector that is meant to facilitate the extraction of excitation energies, but is not experimentally observable. Therefore, I don't think the comparison here is very meaningful, and the authors may consider rephrasing their wording.

The authors chose the Gaussian function for the probability distribution, and mentioned other functions can be chosen, but there is no guideline on what functions would be appropriate. It seems to me that any function whose limit is the delta function would be okay, e.g., Gaussian function in the limit of infinitely small standard deviation (or equivalently infinitely large τ in the manuscript) approaches the delta function. Also, e.g., eq (A6) is a natural property of the delta function. Could the authors comment on this? It is possible that other probability distributions might lead to more efficient sampling, reducing the number of quantum measurements needed.

With NISQ devices, Trotterization only gives reasonably accurate dynamics for very short time, so it would be instructive to show more time-domain results (now only Fig. 2(d) shows time dependent errors, but showing the actual quantities would be helpful), perhaps in the SI.

The equation for $S(\omega)$ on page 2 seems to miss the density operator.

Is the (i,j) in Fig. 1 supposed to be (n,n') ?

Reviewer #2

(Remarks to the Author)

Referee report of "Probing spectral features of quantum many-body systems with quantum simulators"

This paper discusses the novel method for spectroscopy, which only uses a time evolution with a target Hamiltonian with randomly sampled evolution time. The unbiased estimator of spectral detector function shown in Eq. (3) is constructed from

the expectation value of an observable for the time-evolved state. While many of the recently proposed spectroscopy quantum algorithms rely on controlled time-evolution operations, the proposed method circumvents this but requires sufficiently large state-and-observable dependent coherence. This method only necessitates a time evolution duration logarithmic with the required accuracy, hence robust to noise. The authors demonstrated their method with a 13-qubit quantum device.

I think this method is very interesting, due to the hardware-efficiency mentioned above. This algorithm can offer the advantage use of near-term quantum devices. However, I am not fully sure this paper satisfies the strict criteria worth publishing in Nature Communications due to the following reasons:

First, I totally agree that this algorithm is quite hardware-efficient compared with recently proposed ones. However, because I am not so familiar with the conventional spectroscopy experiments, I cannot judge how this method outperforms them. While authors mention that the conventional spectroscopy can be applied to only equilibrium state and weak perturbation, I am not fully sure how problematic this is, especially for a large-scale analog simulators. I hope the authors will have a thorough argument about this.

Second, because the point that only short-time evolution is necessary is a strong point, I suggest that the authors compute the running time complexity (maximal evolution time \times sampling numbers) in the presence of a simple noise model, e.g., global depolarizing noise model, and compare that with the other spectroscopy algorithms. Perhaps, in the presence of noise, the scaling becomes comparable or better than the quantum algorithms achieving Heisenberg scaling.

Also, in the part for assuring the large coherence, I am a bit confused about the introduced basis $|n\rangle = \gamma_q^\dagger |0\rangle$. This is not generally the eigenstate of the Hamiltonian, so I guess authors cannot directly apply this basis to derive Eqs. (1, 2, 3).

The reason I raised the above problems is because the advantages of this method over other methods are a bit unclear to me. I hope the authors will clarify these.

Nevertheless, I think this paper is quite well written and the result is novel, and definitely can be published in high-standard specialized journals such as NPJ quantum information after the suitable modifications.

Reviewer #3

(Remarks to the Author)

This manuscript introduces a novel technique to retrieve the excitation spectrum of a quantum many-body system using a quantum simulator. The protocol involves measuring the dynamics of an observable over suitably chosen time intervals and initial states, thereby eliminating the need for ancillary systems, which are typically required in alternative methods. The authors enhance their study with an analysis of the robustness of their method to noise, as well as numerical simulations for a variety of typical quantum systems, including a simulation of a spin-lattice model on an IBM quantum computer.

The manuscript is well-written, and the analysis is sound. The topic is relevant, timely, and suitable, at least in principle, for the audience of Nature Communications, as evidenced by the recent publication in the journal of other works on the same topic, such as:

[1] Efehan Kökcü, Heba A. Labib, J. K. Freericks, and A. F. Kemper, Nature Communications, volume 15, 3881 (2024).

My main concern regards the significance of the method in comparison to existing techniques. While the novelty of the scheme is apparent, the absence of a dedicated comparative analysis makes it difficult to assess the extent to which the method outperforms existing techniques. The scheme seems similar to methods based on extracting two-time correlation functions within a linear response framework. For instance, the preparation of the initial state in the proposed method could be likened to the perturbation required in these traditional methods, and the measurement of the two-time correlation functions to the measurement of the observable for different time intervals in the method introduced here. Although the authors point out that measuring the correlation function typically requires the use of an ancillary qubit, which their method avoids, it is unclear if this is the only advantage. Using a single ancillary qubit does not seem to be a significant overhead.

In summary, it would be beneficial for the authors to clearly state how their method outperforms methods like that in [1] and others. A detailed comparative analysis would significantly enhance the manuscript's impact and clarify the advantages of the proposed technique.

Version 1:

Reviewer comments:

Reviewer #1

(Remarks to the Author)

The authors have addressed my concerns raised in my previous review, and I appreciate their efforts in substantially

improving the manuscript by adding more resource analysis and more numerical results.

Reviewer #2

(Remarks to the Author)

The authors clearly answered my comments. In particular, polynomial dependence on accuracy, even in the presence of noise, is a critically strong point in this work over existing works. The numerics clearly show the advantages as well. So, I judge this work satisfies the criteria of nature communications and recommend its publication.

Reviewer #3

(Remarks to the Author)

I have carefully reviewed the authors' responses to my comments and those of the other two referees. The authors have thoroughly addressed all the raised questions, and the manuscript has been revised accordingly. These changes have significantly enhanced its clarity. Additionally, the revised version better highlights the relevance of the work in comparison to existing methods.

In my initial report, I had expressed my inclination to recommend this work for publication in Nature Communications. After reviewing the revisions, I remain confident in the value of the work and believe the manuscript is now ready for publication without further modifications. I would like to thank the authors for their thoughtful and detailed responses to my comments.

Reply to Referee 1:

The manuscript by Sun and co-workers proposed an approach that relies on sampling time intervals based on a given probability distribution and the evaluation of the time evolution operator at these time points using quantum computers. The idea of reframing the integration over time to some statistical averaging is interesting in the context of quantum computing (not so new in the context of classical simulations). Requiring only relatively short real-time evolution via Trotterization does not require ancilla qubits, providing advantages over some other quantum algorithms (but not all). The new approach can be useful for many systems, but not all, given its requirement for non-vanishing coherence. Whether the novelty of this work meets the standard of nature communications is up to the editor, but I have a few comments that I think need to be addressed before publication.

Reply: We appreciate the Referee for taking the time to review our work and for the constructive comments. We thank the summary of our work and we are glad to see the positive comments.

We also understand the Referee’s concerns about whether our work shows significance over other existing works. The Referee raised two points in this summary: (1) the advantages of our method compared to others and (2) the conditions for applicability. While the Referee did not request us to address these aspects (indeed, the advantages and applicability of our work are appreciated), we would like to emphasize our unique contributions in relation to these two aspects, which will highlight the significance of our work. Regarding the comparison with other works, our method shows advantages in circuit complexity as the referee pointed out. In the revised manuscript, we further show that our method can achieve polynomial total running time (circuit depth \times sampling numbers) when noise is present. This clearly shows advantages over the existing works, such as quantum phase estimation (QPE) and the one based on the Fourier transform of the time signal, which all have exponential total running time. Regarding the applicability, particularly in relation to the coherence requirement, we agree with the Referee’s comment, but on top of it, we further explore the initial state requirements for lattice problems. In the revised manuscript, we have expanded the discussion on coherence, particularly in the context of representative many-body quantum systems (e.g. Fermi liquid and some non-Fermi-liquid systems), to make this aspect clearer. Therefore, to a great extent, our work overcomes some critical challenges present in existing methods, which underscores its significance and suitability for publication. Below, we elaborate on the two points.

Firstly, we discuss the advantage in resource requirement in reply to the comment on “Requiring only relatively short real-time evolution via Trotterization does not require ancilla qubits, providing advantages over some other quantum algorithms (but not all)”. As existing algorithms focus on different tasks, we agree that it is hard to compare them at the same level. Nevertheless, as the Referee pointed out, the feature of short-time complexity provides advantages over the existing schemes, either by simulation of spectroscopy experiments or quantum computing algorithms using phase estimation. Beyond the advantage in time complexity, in the revised manuscript we find that when device noise is present, our method shows clear advantages over others on the sampling numbers (and thus shows an advantage in the total running time). To be more specific, we find that under noise, the existing methods, such as QPE and spectrum estimation method by Fourier transform of Delta function, require the sampling numbers to be exponential in precision, making these methods exponentially costly in the resource requirement. In contrast, the total running time using our method still maintains polynomial scaling. The reason why other methods are costly in terms of the sample complexity can be understood as follows. To mitigate the effect of noise, it usually introduces an overhead in the number of measurements as the variance of the estimator ($\hat{\sigma}$

and $\hat{G}(\omega)$) will be amplified due to error mitigation. The existing methods require a circuit depth at least linear in inverse precision. Thus, due to the variance issue (e.g. overhead by error mitigation), the total sample complexity will be amplified as $\mathcal{O}(\exp(1/\varepsilon))$. Our result is the first one that achieves a logarithmic growth in circuit depth and a polynomial growth in the total running time.

In the revised manuscript, we added discussions on noise analysis, including resource requirement in the presence of noise, noise mitigation at the algorithmic level, general noise mitigation strategy, numerical verification of noise models, and simulation of the spectral feature detection. Below, we summarise the theoretical and numerical results when device and statistical noise is concerned:

1. The total running time complexity in the presence of noise is polynomial in order to achieve a given precision $\mathcal{O}(\text{poly}(\varepsilon^{-1}))$, while in contrast, existing works will be $\mathcal{O}(\exp(\varepsilon^{-1}))$.
2. Error mitigation strategy for the global depolarising noise based on fitting in a similar vein to randomised benchmarking, and for general types of noise based on probabilistic error cancellation.
3. Numerical verification of the noise model and the fitting process (emulate the actual quantum circuits with local depolarising noise). The fitting results are shown in Figure 3 in Methods and Figures 6 and 7 in Supplementary Section II.3 (which can be found in Fig. 1 in response to Referee 2’s reply).
4. Numerical simulation of the noisy spectroscopic property estimation up to 11 qubits. Verification of the efficacy of the QEM strategy. The results under different conditions (noisy, ideal and the results after error mitigation) in the time domain and frequency domain are shown in Figure 4 in Methods and Figures 6 and 7 in Supplementary Section II.3 (which can be found in Fig. 2 in response to Referee 2’s reply).

We have added the key results in the main text and Methods and added a new section in Supplementary Section II.

Secondly, we reply to the Referee’s comment on “The new approach can be useful for many systems, but not all, given its requirement for non-vanishing coherence”. We appreciate the comment on the usefulness of our new approach. We agree that it cannot be applied to all systems because the coherence condition is a consequence of the hardness of simulating a many-body system, which indicates that in general this task cannot be solved with a quantum computer. Nonetheless, instead of focusing on this general (and impossible) task, a more physically relevant question is to which extent we can apply this new approach to enhance our understanding of many-body systems. In response to that, in the revised manuscript, we analysed the systems which this method can be applied to and added more discussions on the coherence condition. As discussed in the main text, the systems that host quasiparticles satisfy the condition for a nonvanishing coherence. Taking the Fermi liquid as a simple example, the low-energy eigenstates are labelled by a set of quantum numbers $n_{k,\sigma} = 0, 1$, the occupation numbers and the introduction of interactions will not let the quasiparticle decay into multiple other quasiparticles. More concretely, the overlap between the quasiparticle state $|\psi_k\rangle$ carrying momentum k and the ground state $|\psi_0\rangle$ excited by $\hat{c}_{k,\sigma}^\dagger$, $|\langle\psi_k|\hat{c}_{k,\sigma}^\dagger|\psi_0\rangle|^2$, is not zero in the thermodynamic limit for k near the Fermi surface, which gives a guarantee for the nonzero coherence. The non-metallic states such as anti-ferromagnetic states, are generated due to interactions between electrons and can be understood from this Fermi liquid theory. It is worth noting that finding the occupancy of the original orbitals in the interacting eigenstate $|n'\rangle$, $n_{\text{occ}}(k, \sigma) = \langle n' | \hat{c}_{k,\sigma}^\dagger \hat{c}_{k,\sigma} | n' \rangle$ is still a computationally hard task.

Although our method relies on the coherence between the target eigenstates, it could be useful in identifying certain spectroscopic features of the target system. For example, in the case of non-Fermi liquid, the lifetime of quasiparticles is short and its spectrum is expected to be blurred. This spectroscopic feature indicates the beginning of non-Fermi-liquid behaviour. Understanding this blurred behaviour in nature is an interesting question, and our approach can help reveal it. The simulated spectroscopic features are useful for identifying the transition from a Fermi liquid to a non-Fermi-liquid phase and pinpointing the breakdown of where Fermi liquid theory no longer holds.

Again, we have emphasised in the manuscript that this approach cannot solve general systems and whether this can be used to go beyond classical approaches is not the scope of our work.

We would also like to highlight a few conceptual novelties of this work. Our work proposes an efficient way of simulating physics and could help us in understanding the mechanism of many-body physics, thus bringing new perspectives from two distinct notions. Referee 3 also gave a positive comment on this point showing that our work can be suitable for the audience in Nature Communications. Beyond existing works, we have established a framework for analysing the complexity of spectral property estimation tasks, which can be useful for further investigations on these tasks and many other works can be built on top of it. In addition, our work overcomes one of the limitations in existing spectroscopy simulations as our method is not restricted to analogue quantum simulation: the time evolution can be more general, and the initial state preparation can be realised by a quantum circuit. For example, it can be applied to probe features of BCS types of systems which do not conserve particle numbers. We added more discussions on the preparation of the initial state in Supplementary Section III. Besides, it could be useful in stimulating new quantum computing-based methods for resolving complex many-body systems with different conditions, such as various materials-dependent conditions (e.g. doping levels) and environmental conditions (e.g. pressure and temperature).

To sum up, we believe that our work is novel and meets the high standard of Nature Communications.

The manuscript compares their methods with some other methods where two-point unequal time correction functions are needed for spectral simulations. Since these methods may need ancilla qubits, the proposed method has an advantage. However, this is a bit misleading. Some of these methods were intended to compute actual spectra that can be measured experimentally rather than just the energy spectrum (here, experiment means actual spectroscopic measurement, not quantum simulation on actual quantum computers). On the other hand, the current protocol designed a spectral detector that is meant to facilitate the extraction of excitation energies, but is not experimentally observable. Therefore, I don't think the comparison here is very meaningful, and the authors may consider rephrasing their wording.

Reply: We would like to thank the referee for pointing out the comparisons with other works. We agree that we cannot get the experimental observables, because we did not simulate the whole process. This is mostly because the aim of this work is energy extraction, as the Referee pointed out. The comparison should thus be with quantum simulation protocols that either simulate the process directly or focus on spectral property estimation. But we agree that this might be unclear. Therefore, we have added more restrictions and rephrased the wording in the main text. Moreover, we have clarified the three types of techniques for probing spectral features, including (1) real spectroscopy experiments, (2) direct simulation of spectroscopy experiments using quantum simulators, and (3) spectroscopy-inspired quantum simulations. For the task of spectral property estimation,

our method shows advantages over the quantum simulation protocols as it directly processes the spectrum and has features of easy implementation and low computational complexity. We have added the comparisons in Supplementary Section III.

The authors chose the Gaussian function for the probability distribution, and mentioned other functions can be chosen, but there is no guideline on what functions would be appropriate. It seems to me that any function whose limit is the delta function would be okay, e.g., Gaussian function in the limit of infinitely small standard deviation (or equivalently infinitely large tau in the manuscript) approaches the delta function. Also, e.g., eq (A6) is a natural property of the delta function. Could the authors comment on this? It is possible that other probability distributions might lead to more efficient sampling, reducing the number of quantum measurements needed.

Reply: We would like to thank the referee for pointing out the selection of the filtering function. As can be found in the analysis of quantum resources, we divide the error into two parts: error in finite τ and finite truncation T . If we choose the Delta function, which is a very sharp function, using continuous functions to approximate this sharp function will involve a large truncation in x_c . Usually, it is more costly to approximate the Delta function by using real-time dynamics, e^{-ixH} with time length t . People may think about using other functions to approximate it like Chebyshev functions, but they are not native operations by a quantum computer.

We would like to give some comments about the selection of a Gaussian function. Regarding the circuit complexity at each run, using a Gaussian function will result in a logarithmic dependence on the target precision and thus is asymptotically very optimal. By doing so, we can exploit its nice property in short x_c such that the required time complexity is $\mathcal{O}(\log(1/\varepsilon))$.

Different functions will result in different time and sample complexity. For example, another common choice is to think about using a double-side imaginary time evolution as a spectral filter, $p(\omega) = e^{-\beta|\omega|}$. Then by analysing the truncation error, it is not hard to see that x_c is proportional to $1/\varepsilon$, more specifically, it should be $x_c = 2/(\pi\varepsilon)$. This work established a framework for analysing the performance of different methods in this property estimation task.

With NISQ devices, Trotterization only gives reasonably accurate dynamics for very short time, so it would be instructive to show more time-domain results (now only Fig. 2(d) shows time dependent errors, but showing the actual quantities would be helpful), perhaps in the SI.

Reply: We would like to thank the Referee for this suggestion. As discussed above, we find that the time lengths for estimating the spectroscopic features may not be very long: in theory the required time complexity of our method is shorter than other existing methods. Therefore, it is particularly beneficial when running it with NISQ devices. We also numerically verified this point for spin Hamiltonians. For example, in Fig 2 (g), we studied the estimation error with different time lengths (and different system sizes). Figure 2 (g) shows that to ensure the simulation error is less than a threshold, say 0.01, the maximum time length is 6, and it is nearly independent of the system size. With this short time evolution, the error in the time domain can be quite small. As shown in theory and numerical results in Refs. [1, 2], the Trotter error can be quite loose. For example, the actual Trotter error is two orders smaller than the counting bound for the 50-site Heisenberg model (e.g. Fig 3 in Ref. [2]), which provides concrete evidence for the performance of Trotterisation.

Our numerical results also indicate that Trotterisation performs quite well even when the time step is relatively large. In Methods, we considered a more realistic setup by including both gate noise

and measurement noise in the simulation. In this setup, this usually poses challenges for most of the quantum simulation or computing algorithms because of noise. The results show that our method remains effective for a relatively large system size of up to 11 qubits (as simulating the density matrix is usually hard for a laptop). As suggested by the Referee, we present more simulation results in the time domain in Methods and Supplementary Section II. The simulation results with different noise rates and system sizes can be found in Fig. 4(b), Fig. 6(b) and Fig. 7(c).

The equation for $S(w)$ on page 2 seems to miss the density operator. Is the (i,j) in Fig. 1 supposed to be (n,n') ?

Reply: We would like to thank the referee for the careful reading of the notation used in our manuscript. We have fixed these typos. We also checked our manuscript thoroughly to minimise these typos.

Reply to Referee 2:

This paper discusses the novel method for spectroscopy, which only uses a time evolution with a target Hamiltonian with randomly sampled evolution time. The unbiased estimator of spectral detector function shown in Eq. (3) is constructed from the expectation value of an observable for the time-evolved state. While many of the recently proposed spectroscopy quantum algorithms rely on controlled time-evolution operations, the proposed method circumvents this but requires sufficiently large state-and-observable dependent coherence. This method only necessitates a time evolution duration logarithmic with the required accuracy, hence robust to noise. The authors demonstrated their method with a 13-qubit quantum device.

I think this method is very interesting, due to the hardware-efficiency mentioned above. This algorithm can offer the advantage use of near-term quantum devices. However, I am not fully sure this paper satisfies the strict criteria worth publishing in Nature Communications due to the following reasons:

Reply: We appreciate the Referee for the thoughtful and helpful report. We are delighted to hear that the Referee finds the paper interesting and novel. Meanwhile, we also see the Referee’s concern about whether this work satisfies the strict criteria of Nature Communications. That is to ask whether this method can be significantly advantageous over other methods. Indeed, the questions the Referee asked were very insightful and constructive. Thanks to the Referee’s valuable suggestions, we have considered and addressed several aspects that were under-explored in the previous manuscript and the existing literature. Particularly, we clarified the advantages over simulation schemes, the advantages when noise is present, the applicability (e.g. the systems to which our approach can be applied) and the spectroscopic indication.

In the revised manuscript, we explicitly point out the advantages over other quantum simulation or computing protocols. Specifically, our method shows advantages in two aspects: the complexity advantages in terms of circuit depth and sampling numbers (with or without device noise), and the spectral feature detection beyond existing analogue quantum simulations. As suggested by the Referee, we give theoretical analysis for the sample complexity when device noise is present, which shows clear advantages over existing algorithms, evidenced by theoretical and numerical results. This gives a direct response to the Referee’s main technical questions regarding the device noise. With these substantial revisions, we are now even more confident in the novelty and significance of our work. We believe that our manuscript now meets the high standards of Nature Communications and is worthy of publication.

First, I totally agree that this algorithm is quite hardware-efficient compared with recently proposed ones. However, because I am not so familiar with the conventional spectroscopy experiments, I cannot judge how this method outperforms them. While authors mention that the conventional spectroscopy can be applied to only equilibrium state and weak perturbation, I am not fully sure how problematic this is, especially for a large-scale analog simulators. I hope the authors will have a thorough argument about this.

Reply: We appreciate the Referee for this constructive comment. As this comment focuses on the comparison with analogue simulation, we concentrate our discussion on the advantages over analogue simulations and respond with a few examples. Regarding the question of how our method outperforms existing approaches, before going into the details, we first clarify the approaches involved in this question. These include (1) real spectroscopy experiments, (2) direct simulation of

spectroscopy experiments using quantum simulators, and (3) spectroscopy-inspired quantum simulations.

The first approach refers to scattering experiments based on real probes and measurements, which inject neutrons into a target sample (like a synthesised single crystal). They are state-of-the-art experimental probe approaches that are widely used to uncover the complex quantum many-body behaviours. Nevertheless, the cost is huge because of the extreme experimental conditions and the high requirements in synthesising pure materials (pure in the sense that there are not many impurities and the interactions types are clear).

Motivated by spectroscopy experiments, recently, there are some works that propose either simulating spectroscopy directly to study quantum many-body behaviours, or spectroscopy-inspired quantum computing approaches. We guess that the Referee is more curious about whether our protocol can be more advantageous than the second and third approaches, which are to simulate spectroscopy by using either digital quantum computers or analogue quantum computers. Below, we will discuss both schemes.

The drawback of directly simulating the scattering process by a quantum computer is obvious because it inherits the limitation of spectroscopy experiments and did not overcome it. As pointed out here, in the spectroscopy experiment or its simulation, the samples to be probed are in their equilibrium state, and thus, the information is restricted to the diagonal form. An advantage of our method is that because our initial state is not a steady state, and thus it can probe the energy difference between different excited states. In terms of implementation, they need one ancilla qubit that controls the rest of the qubits which will require depth $O(n)$ for lattice problems. In contrast, there is no overhead in compiling the non-local gate in our protocol and thus the depth is $O(1)$.

In our work, the transition between different excited states can be probed given that the coherence is nonzero. One contribution of this work is to analyse how to choose the initial state and the observables in a more concrete way. This is rarely discussed in existing works. For example, in a recent paper, they used the ground state for simplicity (it is generally accepted from the complexity conjecture that both the ground state and thermal states are hard to prepare, and that paper mainly focused on the dynamics simulation without discussing the challenges in the state preparation).

Our method is not restricted to the implementation in an analogue quantum simulation way. Indeed, our method is more versatile and can be useful when FTQC is advent. The reason why the method developed in this work can go beyond standard analogue quantum simulation is that we can employ the programmability of a digital quantum computer for more general initial state preparation and time evolution. To see this point concretely, we will give a class of examples of which analogue quantum computers may be hard to probe. Let us start with a model Hamiltonian for superconductivity. Because of the large numbers of particles involved, the fluctuations in the number of Cooper pairs should be small, which suggests a mean-field approximation to the BCS Hamiltonian. The BCS Hamiltonian becomes quadratic, which reads

$$H = \sum_{k,\sigma} \varepsilon_k \hat{c}_{k,\sigma}^\dagger \hat{c}_{k,\sigma} + \sum_k \left(\Delta_k \hat{c}_{k,\uparrow}^\dagger \hat{c}_{-k,\downarrow}^\dagger + \Delta_k^* \hat{c}_{-k,\downarrow} \hat{c}_{k,\uparrow} \right) \quad (1)$$

where the irrelevant constant is removed. Analogue simulators can hardly simulate this type of Hamiltonians (e.g. its time evolution), which do not conserve particle numbers. However, since it is bilinear in terms of creation and annihilation operators, the Hamiltonian can be diagonalized. Specifically, by using a Bogoliubov transformation, this Hamiltonian can be transformed into a

diagonal form $\mathcal{H} = \sum_{k=1\sigma} \omega_k \hat{\gamma}_{k\sigma}^\dagger \hat{\gamma}_{k\sigma}$ where $\hat{\gamma}_{k\sigma}^\dagger$ and $\hat{\gamma}_{k\sigma}$ are a new set of fermionic operators that satisfy the canonical anti-commutation relations, and can be regarded as a rotated basis with respect to the original one. The rotated basis is related to the original basis by the unitary transformation as

$$\hat{\gamma}_j^\dagger = U c_j^\dagger U^\dagger \quad (2)$$

where $j = (k, \sigma)$ and U is a unitary operator which does not conserve particle numbers. As discussed in quantum computing literature by Google's team [3], this unitary operator can be decomposed into local operators and thus can be implemented easily on quantum computers.

For the interacting case, we can prepare the initial state with a single quasiparticle excitation,

$$|\psi_0\rangle = \hat{\gamma}_j^\dagger |\text{vac}\rangle = U \hat{c}_j^\dagger |\text{vac}\rangle. \quad (3)$$

Here, it is worth noting that $\hat{\gamma}$ can be implemented by applying a unitary transformation to the original basis of \hat{c} . The new operators also satisfy the canonical anticommutation relations.

We understand that the Referee also raised the question about how the creation operation in another basis in Eq. (4) can be realised. To prepare the initial state given by Eq. (4), we only need to apply a unitary operator to an easy-to-prepare state. Therefore, this gives a response to the question about how to realise this basis efficiently. A few discussions will be answered under your next comment. We note that Eq. (4) could serve a good initial state when the quasiparticle picture still holds. The Hamiltonian evolution can be realised by using Trotterisation or a random sampling way that is introduced in the main text.

We would like to remark that the advantages in terms of the time complexity are the key contributions which obviously surpass the previous works as acknowledged by the Referee. As the discussion is focused on the comparison with analogue simulation, we will not mention these advantages here.

Second, because the point that only short-time evolution is necessary is a strong point, I suggest that the authors compute the running time complexity (maximal evolution time \times sampling numbers) in the presence of a simple noise model, e.g., global depolarizing noise model, and compare that with the other spectroscopy algorithms. Perhaps, in the presence of noise, the scaling becomes comparable or better than the quantum algorithms achieving Heisenberg scaling.

Reply: We appreciate the Referee for this very helpful and insightful suggestion. In this revised manuscript, we have conducted a theoretical analysis of the resource requirements when noise is concerned. After analysing the global depolarising model, as suggested by the referee, we find that our method shows a clear advantage over other methods, aligning with the Referee's expectation.

To be more specific, we find that under noise, the existing eigenenergy estimation methods, such as QPE and spectrum estimation method by Fourier transform of Delta function, will be exponentially costly in the number of measurements and hence the total running time complexity. In contrast, the total running time of our method maintains polynomial as suggested by the Referee. The reason why other methods are costly in terms of the number of measurements can be understood in the following way. To mitigate the effect of noise, it usually introduces an overhead in the number of measurements (as the variance is amplified by error mitigation). The existing methods require a circuit depth growing at least linear in inverse precision. Thus, due to the variance issue (e.g. the cost of error mitigation), the total complexity will be amplified as $\exp(1/\varepsilon)$. Our result achieves a logarithmic growth of depth in inverse precision and polynomial total time, showing advantages over others.

Figure 1: **Simulation data and exponential fit of the circuit evolution when local depolarising noise and statistical noise are concerned.** Numerical results of the circuit implementation of noisy spectroscopy protocol simulating the Ising model with $h_x = 2$ and $h_z = 0.1$ for different noise strength p . The solid lines represent the best exponential fit for every p . All the regressions have a relative predictive power $R^2 = 0.99$. Figure 3 in Methods.

Figure 2: **Noisy and error mitigated time dynamics of the spectroscopy protocol.** **a** Time evolution of the observable $\langle Y_i \rangle$ at site $i = 1$. The figure shows the results of the noisy, the error mitigated (QEM) and the ideal (noiseless) cases. **b** Root mean square error (RMSE) of the noisy and the error mitigated time evolutions. **c** Error of the noisy and QEM spectrum in the frequency domain. The green line represents the maximum value of the error in the time domain to compare both results. **d** Error of the spectrum in the k -space. Figure 4 in Methods.

We have performed a series of numerical simulations to verify the efficacy of error mitigation and the performance of our method when noise is concerned. Below, we summarise the main results on noise analysis:

1. The total running time complexity in the presence of noise is polynomial in order to achieve a given precision $\mathcal{O}(\text{poly}(\varepsilon^{-1}))$, while in contrast, existing works will be $\mathcal{O}(\exp(\varepsilon^{-1}))$.
2. Error mitigation strategy for the global depolarising noise based on fitting in a similar vein to randomised benchmarking, and for general types of noise based on probabilistic error cancellation.
3. Numerical verification of the noise model and the fitting process (emulate the actual quantum circuits with local depolarising noise). The fitting results are shown in Figure 3 in Methods and Figures 6 and 7 in Supplementary Section II.3 (which can be found in Fig. 1).
4. Numerical simulation of the noisy spectroscopic property estimation up to 11 qubits (nearly reaching the limit of classical simulation as it involves density matrix simulation). Verification of the efficacy of the QEM strategy. The results under different conditions (noisy, ideal and the results after error mitigation) in the time domain and frequency domain are shown in Figure 4 in Methods and Figures 6 and 7 in Supplementary Section II.3 (which can be found in Fig. 2).

To support Points 3 and 4, we present a few representative numerical results here. In Fig. 1, we numerically verify the behaviour of a local depolarising noise which is applied after each individual gate and find that it can be fitted with an exponentially decaying function. With this result, we perform error mitigation to the noisy measurement outcomes and present the results in the time-domain and frequency domain in Fig. 2.

Next, we test the performance of the simple error mitigation strategy. We consider a more practical setup by choosing a local depolarising noise model. Specifically, we apply local depolarising noise after each gate (including both single-qubit gates and two-qubit gates). We numerically verify the behaviour of the noise model by emulating the actual quantum circuits with local depolarising noise. In Fig. 1, we numerically verify the behaviour of a local depolarising noise which is applied after each individual gate and find that it can be fitted with an exponentially decaying function. The fitting results for the average over all the qubits are shown in Fig. 1. These results indicate that the global white noise could be a good noise ansatz for realistic noise. We include the fittings results for individual observable $\langle Y_i \rangle$ in Supplementary Section II.1.

For the noisy simulation and the error mitigation numerical study, we consider the 1D Ising model from before with an extra term h_z (this model in general is non-integrable). With the fitting result in Fig. 1, we perform error mitigation to the noisy measurement outcomes. The results under different conditions (noisy, ideal and the results after error mitigation) in the time domain and frequency domains are shown in Fig. 2. We can observe that QEM works considerably well in the time domain. As predicted, the algorithm is resilient to noise so we can still recover the spectrum after a noisy evolution. This is why the error is considerably smaller in the frequency (and k-space) domain. We scaled up the simulation up to 11 qubits and the results remain nearly the same. See Figure 7 in Supplementary Information for the results.

We have added the main theoretical results in the main text and elaborated on the methods and the numerical results in Methods and Supplementary Section II. Again, we appreciate the referee

for this insightful suggestion. This noise analysis strengthens the significance of the result.

Also, in the part for assuring the large coherence, I am a bit confused about the introduced basis $|n\rangle = \gamma_{\mathbf{q}}^{\dagger}|0\rangle$. This is not generally the eigenstate of the Hamiltonian, so I guess authors cannot directly apply this basis to derive Eqs. (1, 2, 3).

Reply: We would like to thank the Referee for this comment on the introduced basis and the corresponding coherence.

Firstly, we agree with the referee that the introduced basis is not generally the eigenstate of the Hamiltonian. Nonetheless, regarding the specific confusion of the referee, we first clarify a few things. In deriving Eqs (1,2,3), the eigenbasis of the Hamiltonian, $|n\rangle$ is represented in a general way, which we have not assumed its explicit form. At these places (in Eqs (1,2,3)), we just assumed a nonvanishing coherence, and by doing so we can analyse the resource requirement for ensuring a certain accuracy without knowing the details of the target system.

The mention of the explicit form of $|n\rangle$ is because we wish to look for ways to design the initial state of some typical quantum systems. For example, when the system has well-defined quasi-particles, which is the case for Fermi liquids, an excited state can be understood as a single particle (or quasiparticle) excitation. Therefore, the eigenstate can be represented as $|n\rangle = \gamma_{\mathbf{q}}^{\dagger}|0\rangle$ where $\gamma_{\mathbf{q}}^{\dagger}$ is the rotated creation operation (similar to the case discussed in Eq. (2)). The implementation of this operator has been discussed in the reply to the previous comment.

In addition, we have emphasised in this paper that how to guarantee nonzero coherence is a challenging question and, in general, should be computationally hard; otherwise, it violates the complexity conjecture. Although this is generally believed to be an open challenge and it's not the full scope of this work, we wish to give some comments on the extension of this question. A follow-up question is: what is the condition such that the introduced basis can be written in this way. This holds in many condensed matter systems, like Fermi liquids and the BCS type of Hamiltonians, while a counter-example is spin liquids, which do not have quasiparticles. Taking the Fermi liquid as a simple example, the low-energy eigenstates are labelled by a set of quantum numbers $n_{k,\sigma} = 0, 1$, the occupation numbers and the introduction of interactions will not let the quasiparticle decay into multiple other quasiparticles. More concretely, the overlap between the quasiparticle state $|\psi_k\rangle$ carrying momentum k and the ground state $|\psi_0\rangle$ excited by $\hat{c}_{k,\sigma}^{\dagger}$, $|\langle\psi_k|\hat{c}_{k,\sigma}^{\dagger}|\psi_0\rangle|^2$, is not zero in the thermodynamic limit for k near the Fermi surface, which gives a guarantee for the nonzero coherence. The non-metallic states such as anti-ferromagnetic states, are generated due to interactions between electrons and can be understood from this Fermi liquid theory. Nevertheless, it is worth noting that finding the occupancy of the original orbitals in the interacting eigenstate $|n'\rangle$, $n_{\text{occ}}(k, \sigma) = \langle n'|\hat{c}_{k,\sigma}^{\dagger}\hat{c}_{k,\sigma}|n'\rangle$ is still a computationally hard task.

Although our method relies on the coherence between the target eigenstates, it could be useful in identifying certain spectroscopic features of the target system. For example, in the case of non-Fermi liquids, the lifetime of quasiparticles is short and its spectrum is expected to be blurred. This spectroscopic feature indicates the beginning of non-Fermi-liquid behaviour. Understanding this blurred behaviour in nature is an interesting question, and our approach can help reveal it. The simulated spectroscopic features could be useful for identifying the transition from a Fermi liquid to a non-Fermi-liquid phase and pinpointing the breakdown of where Fermi liquid theory no longer holds. We also would like to remark that the discussion on this point is rarely discussed in previous papers and even in the latest work is not discussed. We have added more physical intuitions on this

point and made it clearer in the revised manuscript.

The reason I raised the above problems is because the advantages of this method over other methods are a bit unclear to me. I hope the authors will clarify these. Nevertheless, I think this paper is quite well written and the result is novel, and definitely can be published in high-standard specialized journals such as NPJ quantum information after the suitable modifications.

Reply: We would like to thank the referee for this comment. In response, we clarified the advantages in the main text. In the revised manuscript, beyond the previously submitted version, we further explore the advantages on the quantum resource and its spectroscopic feature detection sides, which are summarised as follows

1. We have proven that the circuit complexity is near optimal for the spectral property estimation task. We further prove that in the presence of noise, our method is advantageous over all the existing methods, such as quantum computing algorithms by quantum phase estimation or taking a Fourier transform of the time-dependent signals, and quantum simulation of the spectrum directly. We also numerically verified the performance when considering gate and statistical noise.
2. Our work extends beyond existing spectroscopy simulation methods because it can be applied to probe features of systems without particle conservation. It is not restricted to analogue quantum simulation and shows an advantage over it because the time evolution can be more general, and the initial state preparation can be realised by a quantum circuit.
3. We analyse the conditions where the spectroscopic features can be probed, which include systems that host quasi-particles (they can be classically hard to simulate) like the Fermi liquid and BCS types of Hamiltonian. We clarify the limitation as well as the spectroscopic indication of our protocol, which can be useful for researchers to identify the transitions to non-Fermi-liquid phases.
4. Our method does not need control qubits and thus will not introduce any overhead in circuit compilation. When restricting the qubit connectivity, this feature could bring advantage, for example, in simulating lattice models.

We sincerely appreciate the Referee's comments and helpful input, which enlightened us to address aspects that were under-explored in the previous manuscript and the existing literature. We hope that the above discussions have addressed your major concerns and the revised manuscript has met the standard of Nature Communications. We believe that the revised manuscript will stimulate follow-up research from many active sides, including the scattering spectroscopy, many-body physics and quantum computing sides, and thus is more suitable for the readership of Nature Communications. Therefore, we hope to have your favourite consideration for publication.

Reply to Referee 3:

This manuscript introduces a novel technique to retrieve the excitation spectrum of a quantum many-body system using a quantum simulator. The protocol involves measuring the dynamics of an observable over suitably chosen time intervals and initial states, thereby eliminating the need for ancillary systems, which are typically required in alternative methods. The authors enhance their study with an analysis of the robustness of their method to noise, as well as numerical simulations for a variety of typical quantum systems, including a simulation of a spin-lattice model on an IBM quantum computer.

The manuscript is well-written, and the analysis is sound. The topic is relevant, timely, and suitable, at least in principle, for the audience of Nature Communications, as evidenced by the recent publication in the journal of other works on the same topic, such as: [1] Efehan Kökcü, Heba A. Labib, J. K. Freericks, and A. F. Kemper, Nature Communications, volume 15, 3881 (2024).

Reply: We would like to thank the Referee for the positive comments on our manuscript. We are pleased to see that the Referee pointed out that our manuscript is in principle very relevant for the audience in nature communications. We would also like to thank the referee for bringing this very relevant work (Ref. [4]) to us. In the revised manuscript, we have provided a more detailed comparison between our work and other related works (in particular Ref. [4]).

My main concern regards the significance of the method in comparison to existing techniques. While the novelty of the scheme is apparent, the absence of a dedicated comparative analysis makes it difficult to assess the extent to which the method outperforms existing techniques. The scheme seems similar to methods based on extracting two-time correlation functions within a linear response framework. For instance, the preparation of the initial state in the proposed method could be likened to the perturbation required in these traditional methods, and the measurement of the two-time correlation functions to the measurement of the observable for different time intervals in the method introduced here. Although the authors point out that measuring the correlation function typically requires the use of an ancillary qubit, which their method avoids, it is unclear if this is the only advantage. Using a single ancillary qubit does not seem to be a significant overhead.

Reply: We would like to thank the referee for this constructive comment. We will reply to these comments in several steps. The first comment is about whether this approach is more beneficial than the one based on extracting spectral information from a two-point correlation function. The advantages of our method can be seen in two aspects: the applicability in the task of spectral information extraction and the complexity advantages in the circuit depth and the sampling numbers.

The existing methods based on the two-point correlation function as that in [4] are motivated by scattering spectroscopy experiments. They propose to simulate the spectroscopy experiments directly to study quantum many-body behaviours. The drawback of directly simulating the scattering process by a quantum computer is that it inherits the limitation of spectroscopy experiments and does not overcome it. For example, in the spectroscopy experiment (or its simulation), the samples (e.g. the target material) to be probed are in their equilibrium state, and thus, the information is restricted to the diagonal form. A difference of our method is that our initial state is not a steady state, and thus, it can probe the energy difference between different excited states given a nonzero coherence. One contribution of this work is to analyse how to choose the initial state and the observables in a more concrete way, which is rarely discussed in existing works. For example, in [4], they assumed that the ground state could be prepared as the initial state, although they argued that this is not the scope of their work. As indicated by the complexity conjecture, the ground state

and the thermal state are hard to prepare.

Our method is not restricted to the implementation in an analogue quantum simulation way. In the revised manuscript, we also discussed how our method extends beyond other analogue spectroscopy simulation protocols. The reason is that we can employ the programmability of a quantum computer to prepare the initial state and implement the time evolution under more general types of Hamiltonians. A broad class of examples which analogue quantum simulation finds challenges to simulate, include systems that do not preserve particle numbers. Let us illustrate this point more concretely with a model Hamiltonian for superconductivity. Because of the large numbers of particles involved, the fluctuations in the number of Cooper pairs should be small, which suggests a mean-field approximation to the BCS Hamiltonian. The BCS Hamiltonian becomes quadratic, which reads $H = \sum_{k,\sigma} \varepsilon_k \hat{c}_{k,\sigma}^\dagger \hat{c}_{k,\sigma} + \sum_k \left(\Delta_k \hat{c}_{k,\uparrow}^\dagger \hat{c}_{-k,\downarrow}^\dagger + \Delta_k^* \hat{c}_{-k,\downarrow} \hat{c}_{k,\uparrow} \right)$ where the irrelevant constant is removed. Analogue simulators can hardly simulate this type of Hamiltonians (e.g. its time evolution), which do not conserve particle numbers. However, since it is bilinear in terms of creation and annihilation operators, the Hamiltonian can be diagonalized. Specifically, by using a Bogoliubov transformation, this Hamiltonian can be transformed into a diagonal form $\mathcal{H} = \sum_{k=1\sigma} \omega_k \hat{\gamma}_{k\sigma}^\dagger \hat{\gamma}_{k\sigma}$ where $\hat{\gamma}_{k\sigma}^\dagger$ and $\hat{\gamma}_{k\sigma}$ are a new set of fermionic operators that satisfy the canonical anti-commutation relations, and can be regarded as a rotated basis with respect to the original one. The rotated basis is related to the original basis by the unitary transformation as $\hat{\gamma}_j^\dagger = U \hat{c}_j^\dagger U^\dagger$ where $j = (k, \sigma)$ and U is a unitary operator which does not conserve particle numbers. As discussed in quantum computing literature (see Ref. [3]), this unitary operator can be decomposed into local operators in shallow circuit depth, and thus can be implemented efficiently on quantum computers. For the interacting case, we can prepare the initial state with a single quasiparticle excitation,

$$|\psi_0\rangle = \hat{\gamma}_j^\dagger |\text{vac}\rangle = U \hat{c}_j^\dagger |\text{vac}\rangle. \quad (4)$$

Note that the operator $\hat{\gamma}$ can be implemented by applying a unitary transformation to the original basis. We have added the discussion in Supplementary Section III.

Regarding the comment on whether one ancilla will bring any advantage, we give some comments on that. For example, the previous method will need a controlled qubit that controls all the qubits. In terms of the depth, they may become suboptimal. For example, when using Trotterisation for realising the real-time dynamics of lattice models, the depth may become $\mathcal{O}(n)$ when a control qubit is required. In contrast, our protocol remains $\mathcal{O}(1)$ because they can be implemented in parallel.

Finally, we would like to highlight several unique differences and advantages over the recent work Ref. [4] that the Referee pointed out to us. Ref. [4] presents a way of simulating the correlation functions in a linear response framework. Regarding the simulation of the spectroscopic features, this could be regarded as a special case of our method when taking the filter operator to be an identity. In Supplementary Section I.B, we have extended our method to high-order correlation functions; from there, we can clearly see that [4] is a special case by taking the filtering function to be identity. Note that their results for simulating the dynamics of bosonic and fermionic systems could be employed in our method. Spectroscopic signatures appearing in non-linear response can be found in [5] and it could be an interesting future work.

Moreover, Ref. [4] focused on how to simulate the scattering experiment but did not consider the key component in what we can learn from the spectrum and how to design the protocol itself in order to probe the phenomena. In contrast, we discuss the conditions for designing the initial state and the complexity in exploring the spectral features. We show explicitly how to use the properties

of symmetry or the feature of quasiparticle conservation to find the initial state. In the revised manuscript, we have added more discussions about this point. Taking the Fermi liquid as a simple example, the overlap between the quasiparticle state $|\psi_k\rangle$ carrying momentum k and the ground state $|\psi_0\rangle$ excited by $\hat{c}_{k,\sigma}^\dagger$, $|\langle\psi_k|\hat{c}_{k,\sigma}^\dagger|\psi_0\rangle|^2$, is not zero in the thermodynamic limit for k near the Fermi surface, which gives a guarantee for the nonzero coherence. The non-metallic states such as anti-ferromagnetic states, are generated due to interactions between electrons and can be understood from this Fermi liquid theory. It is worth noting that finding the occupancy of the original orbitals in the interacting eigenstate $|n'\rangle$, $n_{\text{occ}}(k, \sigma) = \langle n' | \hat{c}_{k,\sigma}^\dagger \hat{c}_{k,\sigma} | n' \rangle$ is still a computationally hard task.

Although our method relies on the coherence between the target eigenstates, it could be useful in identifying certain spectroscopic features of the target system. For example, in the case of non-Fermi liquid, the lifetime of quasiparticles is short and its spectrum is expected to be blurred. This spectroscopic feature indicates the beginning of non-Fermi-liquid behaviour. Understanding this blurred behaviour in nature is an interesting question, and our approach can help reveal it. The simulated spectroscopic features are useful for identifying the transition from a Fermi liquid to a non-Fermi-liquid phase and pinpointing the breakdown of where Fermi liquid theory no longer holds. Again, we have emphasised in the manuscript that this approach cannot solve general systems and whether this can be used to go beyond classical approaches is not the scope of our work. The discussions on how to overcome the limitations set up by the initial state and the observables could stimulate other follow-up works.

In the previous works, the key metrics in relation to the required time complexity and sample complexity are not discussed. In this work, we identified the key components involved in this task and highlighted the complexity dependence on these key parameters. We derive the time complexity and the sampling complexity in order to achieve a certain simulation accuracy. The complexity results show clear advantages over the method by taking a Fourier transform of the time-dependent signals. Moreover, our work establishes a framework for analysing the resource cost required for spectral property estimation tasks, which makes the comparison with other similar works easier.

To summarise, the key advantages include:

1. In the presence of noise, our method is advantageous over all the existing methods, such as quantum computing algorithms by quantum phase estimation or taking a Fourier transform of the time-dependent signals, and quantum simulation of the spectrum directly. In the revised manuscript, we added the new results in the main text, Methods and a new section (Supplementary Section II) in SI. These include a theoretical analysis of the resources required when incorporating noise and noise mitigation strategies for simple noise. We also support the theoretical findings with a series of numerical simulations, where we verify the efficacy of noise mitigation and show results for spectral property estimation when device and statistical noise are concerned.
2. Our method does not need control qubits and thus will not introduce any overhead in circuit compilation. When restricting the qubit connectivity, this feature could bring an advantage in circuit depth, for example, when simulating lattice models.
3. Our work extends beyond existing spectroscopy simulation methods because it can be applied to probe features of systems without particle conservation. It is not restricted to analogue quantum simulation and shows an advantage over it because the time evolution can be more general, and the initial state preparation can be realised by a quantum circuit.

4. We have analysed the conditions where the spectroscopic features can be probed. We have added more discussions about the spectroscopic indications (e.g. identifying the Fermi-liquid features and the transitions to non-Fermi-liquid phases) in the main text and Methods.

We also summarise the new theoretical and numerical results in relation to Point 1 when noise is concerned:

1. The total running time complexity in the presence of noise is polynomial in order to achieve a given precision $\mathcal{O}(\text{poly}(\varepsilon^{-1}))$, while in contrast, existing works will be $\mathcal{O}(\exp(\varepsilon^{-1}))$.
2. Error mitigation strategy for the global depolarising noise based on fitting in a similar vein to randomised benchmarking, and for general types of noise based on probabilistic error cancellation.
3. Numerical verification of the noise model and the fitting process (emulate the actual quantum circuits with local depolarising noise). The fitting results are shown in Figure 3 in Methods and Figures 6 and 7 in Supplementary Section II.3 (which can be found in Fig. 1 in response to Referee 2's reply).
4. Numerical simulation of the noisy spectroscopic property estimation up to 11 qubits (nearly reaching the limit of classical simulation as it involves density matrix simulation). Verification of the efficacy of the QEM strategy. The results under different conditions (noisy, ideal and the results after error mitigation) in the time domain and frequency domain are shown in Figure 4 in Methods and Figures 6 and 7 in Supplementary Section II.3 (which can be found in Fig. 2 in response to Referee 2's reply).

We have added the key results in the main text and Methods and added a new section in Supplementary Section II.

We would like to thank the Referee for the comments on the advantages of our work over the existing works. This gives us an opportunity to better present our results and make the results much clearer.

In summary, it would be beneficial for the authors to clearly state how their method outperforms methods like that in [1] and others. A detailed comparative analysis would significantly enhance the manuscript's impact and clarify the advantages of the proposed technique.

Reply: We appreciate the Referee for pointing out Ref. [4]. As discussed above and can be found in the revised manuscript, we clarified the comparisons with Ref. [4], as well as other related works (in different aspects and domains). We also addressed the aspects that were under-explored in the previous manuscript and the existing literature, such as exploring the advantages over analogue simulation, the scaling advantage when noise is present, and the spectroscopic indication.

The Referee's comments greatly helped strengthen our work. We hope that our manuscript meets the high standard of Nature Communications.

References

- [1] Andrew M Childs, Dmitri Maslov, Yunseong Nam, Neil J Ross, and Yuan Su. Toward the first quantum simulation with quantum speedup. *Proceedings of the National Academy of Sciences*, 115(38):9456–9461, 2018.

- [2] Andrew M Childs, Yuan Su, Minh C Tran, Nathan Wiebe, and Shuchen Zhu. Theory of trotter error with commutator scaling. *Physical Review X*, 11(1):011020, 2021.
- [3] Zhang Jiang, Kevin J Sung, Kostyantyn Kechedzhi, Vadim N Smelyanskiy, and Sergio Boixo. Quantum algorithms to simulate many-body physics of correlated fermions. *Physical Review Applied*, 9(4):044036, 2018.
- [4] Efehan Kökcü, Heba A Labib, JK Freericks, and AF Kemper. A linear response framework for quantum simulation of bosonic and fermionic correlation functions. *Nature Communications*, 15(1):3881, 2024.
- [5] Rahul M Nandkishore, Wonjune Choi, and Yong Baek Kim. Spectroscopic fingerprints of gapped quantum spin liquids, both conventional and fractonic. *Physical Review Research*, 3(1):013254, 2021.